# Physics-Preserving Compression of High-Dimensional Plasma Turbulence Simulations

## Abstract

High-fidelity scientific simulations are now producing unprecedented amounts of data, creating a storage and analysis bottleneck. A single simulation can generate tremendous data volumes, often forcing researchers to discard valuable information. A prime example of this is plasma turbulence described by the Gyrokinetic equations: nonlinear, multiscale, and 5D in phase space. They represent one of the most computationally demanding frontiers of modern science, with runs taking weeks and resulting in tens of terabytes of data dumps. The increasing storage demands underscore the importance of compression, however, compressed snapshots might not preserve essential physical characteristics after reconstruction. To assess whether such characteristics are captured, we propose a spatiotemporal evaluation pipeline which accounts for structural phenomena and multi-scale transient fluctuations. Indeed, we find that various compression techniques lack preservation of temporal turbulence characteristics. Therefore, we explore Physics-Informed Neural Compression (PINC), which incorporates physics-informed losses tailored to gyrokinetics and enables extreme compressions of over $100{,}000\times$. This direction provides a viable and scalable solution to the prohibitive storage demands of gyrokinetics, enabling post-hoc analyses that were previously infeasible.

## 1 Introduction

Scientific computing is on the cusp of entering an era of high-fidelity simulations across various domains, such as plasma physics (Fedeli et al., 2022; Chang et al., 2024; Dominski et al., 2024; Kelling et al., 2025), weather and climate modelling (Govett et al., 2024; Bodnar et al., 2024), astrophysics (Grete et al., 2025), and beyond. This progress is driven by advancements in High-Performance Computing (HPC): hardware accelerators, exascale computing systems, and scalable numerical solvers are pushing the horizon of what can be computed. These developments allow practitioners to move beyond reduced numerical approaches and attempt high-fidelity simulations, which are essential to accurately capture the underlying physics of complex systems. A striking instance of such simulations is gyrokinetics (Frieman & Chen, 1982; Krommes, 2012; Peeters et al., 2009), a five-dimensional (5D) nonlinear system that simulates turbulence in magnetised plasmas, such as those found in magnetically-confined nuclear fusion devices.

Gyrokinetic simulations generate massive data volumes that create a severe storage and analysis bottleneck. This arises from their 5D nature, combined with the high-resolution needed to model plasma turbulence. The gyrokinetic equations express the time evolution of particles in a plasma via a 5D distribution function $\boldsymbol{f} \in \mathbb{C}^{v_\parallel \times \mu \times s \times x \times y}$, with spatial coordinates $x$, $y$, $s$ and velocity-space coordinates $v_\parallel$, $\mu$. A single run can produce tens of terabytes of data with snapshots saved at many time steps. In practice, researchers only store diagnostics, making comprehensive post-hoc analysis impossible. Compression offers a remedy by reducing the cost of storing full 5D fields. However, no evaluation framework currently exists to assess whether compressed snapshots preserve transient turbulence dynamics, an essential requirement for post-hoc analysis.

As a solution, we introduce an evaluation framework for transient turbulence characteristics in compressed snapshots of gyrokinetic simulations. To this end, we disentangle *transient fluctuations*, which capture energy transfer across time, from *spatial* quantities, which describe the properties of a single snapshot. We find that various compression techniques fail to preserve transient turbulence

properties. To this end, we explore PINC for turbulent gyrokinetic data. We consider two paradigms: autoencoders (e.g., VQ-VAE (van den Oord et al., 2017)) generalizing on unseen samples, and neural implicit fields (or representations) (Mildenhall et al., 2020; Park et al., 2019), which typically encode individual snapshots into network parameters. Unlike conventional compression, PINC enforces the preservation of key physical quantities, ensuring that downstream scientific analyses remain valid even at extreme compression rates of over $70,000\times$.

We demonstrate that PINC achieves extreme storage reduction while preserving transient turbulence and steady-state spatial characteristics. Both autoencoders and neural fields attain field reconstruction errors comparable to or better than conventional approaches at the same compression rate, while significantly improving physics preservation. A predictable rate-distortion scaling is also observed between compression rate, signal reconstruction and physics fidelity, allowing this trade-off to be estimated a priori. Lastly, we showcase some additional weight space experiments, further pushing the compression levels. Our framework enables detailed analysis of gyrokinetic simulations at scales previously impractical. In summary, we make the following contributions: ❶ we present a spatiotemporal evaluation pipeline to assess physics preservation. It accounts for both spatial structural information and temporal dynamics, together capturing multi-scale transient fluctuations prevalent in turbulent dynamics, and ❷ we introduce a novel physics-informed training curricula for neural compression, PINC in short, equipping different techniques with *gyrokinetics-specific* physical losses, capturing both essential integrals and turbulence characteristics.

## 2 RELATED WORK

**Compression** of spatiotemporal data is not a novel topic, and fields such as numerics and HPC conducted a great deal of research in this direction (Diffenderfer et al., 2019; Lakshminarasimhan et al., 2011; Lindstrom, 2014; Ballester-Ripoll et al., 2019; Momenifar et al., 2022). Related research exists in the domain of computational plasma physics (Anirudh et al., 2023), in particular for Particle-In-Cell (PIC) simulations (Birdsall & Langdon, 2005; Tskhakaya, 2008). The most relevant works include ISABELA (Lakshminarasimhan et al., 2011), an advanced spline method that promises almost lossless compression of spatiotemporal data of up to $7\times$; and VAPOR (Choi et al., 2021), a deep learning method based on autoencoders **[QH9G]** that compresses 2D PIC velocity space slices, supervised with mass, energy and momentum conservation losses. Concurrent work Kelling et al. (2025) proposes streaming pipelines for petascale PIC simulations, learning from data *in-transit* without intermediate storage. While PIC resolves the full 6D plasma kinetics, gyrokinetics reduces the problem to 5D by averaging over fast gyromotion, enabling turbulent simulations too complex for PIC. Beyond compression methods, a closely related line of work is super-resolution (SR), which seeks to reconstruct high-resolution fields from coarsened inputs (Fukami et al., 2023; Yang et al., 2025; Page, 2025). We address the complementary challenge of compactly storing full snapshots.

**Implicit Neural Fields** encode information in a compact feature space, enabling scalable, grid-agnostic representation of high-resolution data. They represent continuous signals as coordinate-based learnable functions (Mildenhall et al., 2020; Park et al., 2019; Dupont et al., 2022a; Mescheder et al., 2019). In general, neural fields map input coordinates to the respective values of a field, i.e. $f_\theta : \mathbb{R}^d \to \mathbb{R}^n$ (Xie et al., 2021). They are usually implemented as MLPs with special activation functions (Sitzmann et al., 2020; Saragadam et al., 2023; Elfwing et al., 2017). In physics, neural fields have been applied to time-varying volumetric data compression (Han et al., 2024) and spatiotemporal dynamics forecasting using implicit frameworks (Serrano et al., 2023), among others.

**Physics-Informed Neural Networks** (PINNs) combine neural networks with physical constraints originating from mathematical formulations (Karniadakis et al., 2021). This is typically done by including additional loss terms (Raissi et al., 2019; Cai et al., 2021), ensuring that the laws of physics are obeyed. Physical constraints such as boundary conditions and conservation laws (Baez et al., 2024) are respected in the learned solutions, and more generally that neural network outputs remain consistent with the underlying differential equations. They have been especially effective in solving forward and inverse partial differential equation problems (Raissi et al., 2019). **[Dmk1, kwUE, WgAS]** Inversly to the typcal local, residual PINN losses, in our case they are global non-linear integrals which depend on the Fourier mode structure. Sitting at the intersection of PINNs and neural compression, Cranganore et al. (2025) combine neural fields with Sobolev training (Son

et al., 2021; Czarnecki et al., 2017) to achieve impressive compression, tensor derivative accuracy and high-fidelity reconstruction on storage intensive general relativity data. **[Dmk1, kwUE, WgAS]** Another notable mention is Momenifar et al. (2022), which uses a physics-informed VQ-VAE to capture velocity gradients and statistical properties in isoentropic flows. Our work systematically evaluates whether compressed representations accurately preserve plasma turbulence-specific quantities, motivating the need for physics-informed loss terms.

## 3 METHODS

### 3.1 EVALUATING PLASMA TURBULENCE

We assess whether compressed representations faithfully capture gyrokinetic turbulence through two complementary groups of metrics, focusing on: **(1)** spatial information, evaluated using non-linear field integrals and turbulence spectra, which measure how well the compressed representations preserve spatial mode structures and energy distributions. **(2)** Temporal consistency, via optical-flow distance and a novel Dynamic Mode Decomposition (DMD) error. These quantify the fidelity of the reconstructed sequence.

**Integrals.** In gyrokinetics, (scalar) heat flux $Q \in \mathbb{R}$ and real-space electrostatic potential $\phi \in \mathbb{C}^{x \times s \times y}$ are two core quantities. They describe essential spatial and physical attributes of the density $\boldsymbol{f}$. $Q$ and $\phi$ are integrals of the distribution function $\boldsymbol{f}$ and are formulated as

$$\phi = \mathbf{A} \int \mathbf{J_0} \boldsymbol{f} \, \mathrm{d}v_\parallel \mathrm{d}\mu, \quad Q = \int \mathbf{B} \int \mathbf{v}^2 \phi \boldsymbol{f} \, \mathrm{d}v_\parallel \mathrm{d}\mu \, \mathrm{d}x \mathrm{d}y \mathrm{d}s, \tag{1}$$

where $\mathbf{A}, \mathbf{B} \in \mathbb{R}^{x \times s \times y}$ encompass geometric and physical parameters, $\mathbf{v} \in \mathbb{R}^{v_\parallel \times \mu}$ is the particle energy, and $\mathbf{J_0}$ denotes the zeroth-order Bessel function. The electrostatic potential $\phi$ is obtained by integrating in the velocity-space from $\boldsymbol{f}$, while the heat flux $Q$ depends on both $\boldsymbol{f}$ and $\phi$. Intuitively, $\phi$ represents the spatial variation of the electric field, while $Q$ measures the energy flow carried by particles along the field lines.

**Wavespace distribution (diagnostics).** Going further, some derived quantities are used by researchers to determine the properties of a simulation and for *diagnosing* the soundness of a given configuration; they measure how energy and electrostatic fluctuations are distributed across modes in wavenumber space, and provide a basis for identifying patterns and behaviors that define turbulent transport in the plasma. In particular, $k_y^{\mathrm{spec}} \in \mathbb{C}^{k_y}$ describes the perpendicular scales of turbulence along $y$, and $Q^{\mathrm{spec}} \in \mathbb{C}^{k_y}$ links turbulent structures to heat transport. They are expressed as convolutions of $\phi$ and $\boldsymbol{Q}$,

$$k_y^{\mathrm{spec}}(y) = \sum_{s,x} |\hat{\phi}(x, s, y)|^2, \quad Q^{\mathrm{spec}}(y) = \sum_{v_\parallel, \mu, s, x} \boldsymbol{Q}(v_\parallel, \mu, s, x, y), \tag{2}$$

where $\hat{\phi}$ is the Fourier space electrostatic potential, and $\boldsymbol{Q}$ is the flux field (also in Fourier space) before applying the outermost integral, which aggregates it to $Q$. Diagnostics are used to characterize turbulence, and can be analyzed both in a time-averaged or transient manner. Time dependency is used to observe how the energy cascade shifts in the energy to lower modes and vice versa, while statistically-steady forms (time-averaged, $\overline{k_y^{\mathrm{spec}}}$ and $\overline{Q^{\mathrm{spec}}}$) define dominant modes. Namely, $\overline{k_y^{\mathrm{spec}}}$ is the mean turbulent spectrum, and $\overline{Q^{\mathrm{spec}}}$ quantifies the heat flux contribution to turbulent transport. They are both used by researchers to detect whether turbulence develops and at which scale.

**[kwUE, QH9G] Time dynamics.** Turbulence is inherently a spatiotemporal phenomena, and a purely spatial evaluation is insufficient to assess reliable reconstruction. To that end, we include metrics from two different perspectives to quantify temporal consistency. First, the fidelity at which the onset of turbulence is reproduced can be assessed in the transitional phase, between the linear and the statistically-steady state of a simulation. We quantitatively evaluate the time-accumulated optimal transport of the wavespace distributions $k_y^{\mathrm{spec}}$ and $Q^{\mathrm{spec}}$ (Equation (2)) through Wasserstein distance (WD). It captures how well the bi-directional energy cascade is captured by the compressed

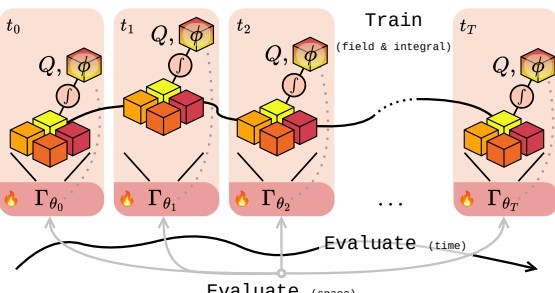

Figure 1: Sketch of the training and evaluation for Physics-Informed Neural Compression (PINC) models. Training is done at individual time snapshots for scalability, while evaluation considers turbulence characteristics, taking both spatial and temporal information into account.

snapshots. Given two sequences of diagnostic pairs $k_y^{\text{spec}}, Q^{\text{spec}}$ and predicted $\widehat{k}_y^{\text{spec}}, \widehat{Q}^{\text{spec}}$ of $N$ subsequent timesteps in the transition phase,

$$\text{EC}_{k_y} = \sum_{i=1}^{N} \text{WD}(k_{y,i}^{\text{spec}}, \widehat{k}_{y,i}^{\text{spec}}), \quad \text{EC}_Q = \sum_{i=1}^{N} \text{WD}(Q_i^{\text{spec}}, \widehat{Q}_i^{\text{spec}}). \tag{3}$$

Second, to check the dynamic consistency of the decompressed sequence we employ the *EndPoint Error* (EPE) of the optical flow field (Baker et al., 2011), commonly used in video modeling (Argaw & Kweon, 2022; Ma et al., 2024). Given two sequences of $x_1$ and $x_2$ of $N$ frames and their $i$-th flow vectors $\mathbf{F}_1^{(i)}$ and $\mathbf{F}_2^{(i)}$, the EndPoint Error is

$$\text{EPE}(x_1, x_2) = \frac{1}{N} \sum_{i=1}^{N} \|\mathbf{F}_1^{(i)} - \mathbf{F}_2^{(i)}\|_2^2. \tag{4}$$

Additional definitions and information can be found in Appendix C.2.

### 3.2 NEURAL COMPRESSION

We identify two dominant approaches to learned compression, depending on a few key aspects. The first approach are autoencoders, with explicit latent space compression at the bottleneck between an encoder and a decoder. Parameters $\theta$ are shared across snapshots and time, and a single model $\Gamma_\theta$ is trained on a dataset. Compression is applied to unseen samples. VQ-VAE (van den Oord et al., 2017) exemplifies autoencoders designed for compression. In contrast, neural implicit representations overfit an independent set of parameters at each datapoint, for instance across time $[\Gamma_{\theta_t}]_{(0...T)}$. Encoding is implicit in weight-space and reconstruction happens by querying the neural field. Figure 1 outlines PINC training and evaluation for a trajectory. The complex Mean Squared Error (cMSE) on the density $\boldsymbol{f}$ is used as reconstruction loss in training

$$\mathcal{L}_{\text{recon}} = \sum_{v_{\parallel}, \mu, x, s, y} \left\| \Re(\boldsymbol{f}_{\text{pred}} - \boldsymbol{f}_{\text{GT}})^2 + \Im(\boldsymbol{f}_{\text{pred}} - \boldsymbol{f}_{\text{GT}})^2 \right\|^2. \tag{5}$$

**5D autoencoders.** Due to the high-dimensional nature of the data, we leverage nD swin layers (Galletti et al., 2025; Paischer et al., 2025a), based on Shifted Window Attention (Liu et al., 2021), which promise scaling to higher dimensions. They work by first partitioning the domain in non-overlapping *windows*, then performing attention only locally within the window. An autoencoder $\Gamma_\theta : \mathbb{C}^{(v_{\parallel}, \mu, s, x, y)} \times \mathbb{R}^4 \to \mathbb{C}^{v_{\parallel}, \mu, s, x, y}$, with $\Gamma_\theta(\boldsymbol{f}, \boldsymbol{c}) = \mathcal{D} \circ \mathcal{E}(\boldsymbol{f}, \boldsymbol{c})$, encodes the 5D density field $\boldsymbol{f}$ and conditioning $\boldsymbol{c}$ containing four gyrokinetic parameters ($R/L_T$, $R/L_n$, $q$, and $\hat{s}$) into a compact latent space, then decodes it to reconstruct $\boldsymbol{f}$. Following hierarchical vision transformers (Liu et al., 2021), the encoder $\mathcal{E}$ tiles $\boldsymbol{f}$ into patches and applies interleaved Swin and downsampling layers. At the bottleneck, channels are downprojected to control the compression rate. The decoder $\mathcal{D}$ mirrors this, with upsampling to restore the original resolution. We apply both regular Autoencoders (AE) and Vector-Quantized Variational Autoencoders (VQ-VAEs) (van den Oord et al., 2017). **[kwUE]**

Autoencoders are *monolithic models* that compress in an explicit latent space, enabling cheap compression and decompression. However, they usually require expensive offline training and a diverse dataset to generalize across different simulations.

**Neural implicit fields.** The distribution function $\boldsymbol{f}$ is indexed by a five-tuple of coordinates $(v_\parallel, \mu, s, x, y)$. Specifically, we train a separate (discrete) coordinate-based Neural Field $\Gamma_{\theta_{t,c}} : \mathbb{N}^5 \to \mathbb{C}$ to fit each $\boldsymbol{f}_t^c$ at time $t$ of a trajectory configured by $c$. Indices are encoded with a learnable embedding hashmap (Müller et al., 2022), then passed to an MLP using SiLU (Elfwing et al., 2017), sine (Sitzmann et al., 2020) or Gabor (Saragadam et al., 2023) activations. Fitting a $\Gamma_{\theta_{t,c}}$ takes ∼1-2 minutes (NVIDIA H100), and since independent networks are used per snapshot training is highly parallelizable or can be performed in a staggered, pipelined fashion for data streams. **[kwUE]** Neural fields are *micromodels*: individual samples are implicitly encoded into network weights, offering resolution invariance and low training requirements. Conversely, encoding is relatively costly.

### 3.3 PHYSICS-INFORMED NEURAL COMPRESSION (PINC)

Training on $\mathcal{L}_{\text{recon}}$ alone cannot ensure conservation of physical quantities or turbulent characteristics. Further, due to the limited representation power, lossy compression inevitably discards useful information if left unconstrained. We supervise on the physical quantities listed in Section 3.1 by penalizing (absolute) deviations from the ground truth. Integral and wavespace losses are defined as

$$\mathcal{L}_Q = |Q_{\text{pred}} - Q_{\text{GT}}|, \qquad \mathcal{L}_\phi = \text{L1}(\phi_{\text{pred}}, \phi_{\text{GT}}),$$
$$\mathcal{L}_{k_y} = \text{L1}(k_{y,\,\text{pred}}^{\text{spec}}, k_{y,\,\text{GT}}^{\text{spec}}), \qquad \mathcal{L}_{Q^{\text{spec}}} = \text{L1}(Q_{\text{pred}}^{\text{spec}}, Q_{\text{GT}}^{\text{spec}}). \tag{6}$$

In addition, we introduce a first-order constraint to capture the turbulent energy cascade. In the case of simulations with a single energy injection scale, the spectra must be monotonically decreasing after the dominant mode, indexed by the peak wavenumber $k_{\text{peak}}$. This specific monotonicity loss can be written as the log-transformed isotonic loss, penalizing negative slopes.

$$\mathcal{L}_{\text{iso}}(k) = \frac{1}{N - k_{peak}} \sum_{k_{\text{peak}}}^{N-1} \left| \log spec(k) - \log spec(k)^{\text{sorted}} \right|. \tag{7}$$

Combining all terms yields the final physics-informed loss:

$$\mathcal{L}_{\text{PINC}} = \underbrace{\mathcal{L}_Q + \mathcal{L}_\phi}_{\mathcal{L}_{\text{int}}} + \underbrace{\mathcal{L}_{k_y^{\text{spec}}} + \mathcal{L}_{Q^{\text{spec}}}}_{\mathcal{L}_{\text{diag}}} + \underbrace{\mathcal{L}_{\text{iso}}(k_{y,\,\text{pred}}^{spec}) + \mathcal{L}_{\text{iso}}(Q_{\text{pred}}^{spec})}_{\mathcal{L}_{\text{grad}}}. \tag{8}$$

**[Dmk1, kwUE, WgAS]** Importantly, our training supervises the model on nonlinear integrals of the distribution function, rather than directly on PDE residuals (Karniadakis et al., 2021) or derivatives (Son et al., 2021). This way PINC implicitly directs the network to the physically relevant modes. In turn, as the $Q$ and $\phi$ integrals depend on the full spectral structure of $\boldsymbol{f}$, many of the losses in Equation (8) are *global* quantities, rather than the local pointwise supervision typical in neural fields and PINNs. $\mathcal{L}_{\text{PINC}}$ can be included in training, but with two caveats: **(i)** loss terms depend on $\boldsymbol{f}$'s mode composition, and **(ii)** global loss terms cannot be computed on coordinate-level. We address **(i)** by applying $\mathcal{L}_{\text{PINC}}$ after $\boldsymbol{f}$'s have converged, to ensure that structure is present. **(ii)** is problematic only for local or sparse methods. The following sections details the tricks req to enable PINC training on neural fields and autoencoders.

**PINC-neural fields.** Neural fields fit $\mathcal{L}_{\text{PINC}}$ continuing optimization after the initial epochs where $\boldsymbol{f}$ is fit. Multi-objective optimizers offer a more principled training stabilization alternative to schedulers or manual learning rate tweaking. Conflict-Free Inverse Gradients (Liu et al., 2024, ConFIG) and Augmented Lagrangian Multipliers (Basir & Senocak, 2023) are commonly employed in PINNs and tasks with many competing losses (Berzins et al., 2025). We focus on ConFIG due to its ease of integration and promising results. Finally, even though neural fields are normally trained on small sparse coordinate batches, $\mathcal{L}_{\text{PINC}}$ gradients can only be computed on the entire grid.

**PINC-autoencoders.** Training autoencoders with physics constraints across heterogeneous samples tends to result in training instabilities; therefore, we employ parameter-efficient fine-tuning to ensure stability. Specifically, we pre-train the autoencoder on $\mathcal{L}_{\text{recon}}$, and finetune it on $\mathcal{L}_{\text{PINC}}$ using Explained Variance Adaptation (Paischer et al., 2025b, EVA), an improved variant of LoRA-style adapters (Hu et al., 2022). For more training details we refer to Appendix C.5.

## 4 RESULTS

The neural fields are simple MLPs with SiLU activations (Elfwing et al., 2017), 64 latent dimension, 5 layers and skip connections. The input matrix locations are encoded with a (discrete) learnable embedding hashmap. Neural fields are fit using AdamW (Loshchilov & Hutter, 2019) with learning rate decaying between $[5e-3, 1e-12]$ (details in Appendix C.6). Results suggest that neural fields trained with ConFIG are less accurate on physical losses, but lead to a marginally better reconstruction error (Appendix Table 4). For simplicity, all neural fields reported are trained with AdamW and no loss balancing, unless specified otherwise. Grid searches and ablations are in Appendix C.6.

As for standard autoencoders and VQ-VAEs, swin tokens are 1024-dimensional, bottleneck dimension is 32, and the codebook dimension of the VQ-VAE is 128, totaling at ∼152M parameters. Both are trained and fine-tuned on 6,890 $f$ time snapshots, amounting to around 500GB of data (details in Appendix B). Compression/reconstruction is subsequently expected to happen *out of distribution*, to unseen trajectories. Pre-training takes ∼60 hours (200 epochs, 4× NVIDIA H100) for standard AE and VQ-VAE. Fine-tuning with EVA weights takes ∼28 hours on one NVIDIA H100 for 120 epochs, adapting ∼4% (6M) of the total parameters. Optimized using Muon (Jordan et al., 2024) with cosine scheduling of the learning rate between $[2e-4, 4e-6]$ (details in Appendix C.5).

We compare with traditional compression based on different techniques: ZFP (Lindstrom, 2014), a very popular compression method for scientific data relying on block-quantization, Wavelet-based compression, spatial PCA and JPEG2000 adapted for the 5D data. Baselines are tuned to achieve compression rates (CRs) of around $1,000\times$ (99.9% size reduction), comparable with neural fields and vanilla autoencoders. For reference, *off-the-shelf* traditional techniques such as `gzip` achieve a lossless compression ratio of ∼1.1x (8% reduction). Information on baselines can be found in Appendix C.3. General and more detailed information about runtime can be found in the Appendix, Table 10. For all visualizations, aspect ratio is set to 2 and does not represent the physical one.

### 4.1 COMPRESSION

We evaluate all methods on traditional compression metrics, integral, and turbulence errors. To measure spatial $f$ reconstruction quality after compression, Peak Signal-to-Noise-Ratio (PSNR) is reported (defined in Appendix C.1). To evaluate temporal compression, we report the EndPoint Error (EPE) (Equation (4)) for turbulent snapshots of $f$. Integral errors are reported as mean absolute error of flux $Q$ and potential $\phi$ after integration of $f$ according to Equation (1). For steady-state turbulence evaluation we normalize the *time-averaged*, $\overline{k_y^{\text{spec}}}$ and $\overline{Q^{\text{spec}}}$ spectra and employ Wasserstein Distance (WD), which is commonly used as a geometry-aware distance metric and can efficiently be computed for 1D spectra. We report additional metrics for spatial evaluation in Table 9. **[kwUE, QH9G]** Furthermore, we provide additional evaluation for transient dynamics in Paragraph 4.2 (Figure 5).

Table 1: Comparison between neural fields, PINC and traditional methods on compression and physical metrics. Evaluation on 60 total $f_t^c$s (10 turbulent trajectories, 6 timesteps), sampled in the statistically steady phase. Errors in data space. Best result in bold, second best underlined.

| | | Compression $f$ | | | Integrals $Q, \phi$ | | Turbulence $Q^{\text{spec}}, k_y^{\text{spec}}$ | |
| --- | --- | --- | --- | --- | --- | --- | --- | --- |
| | CR | L1 ↓ | PSNR ↑ | EPE ↓ | L1(Q) ↓ | PSNR(φ) ↑ | WD($\overline{k_y^{\text{spec}}}$) ↓ | WD($\overline{Q^{\text{spec}}}$) ↓ |
| ZFP | 991× | 0.65 | 28.66 | 0.25 | 87.32 | -16.13 | 0.0228 | 0.0889 |
| Wavelet | 1149× | 0.45 | 32.65 | 0.12 | 86.92 | -13.42 | 0.0228 | 0.0108 |
| PCA | 1020× | 0.47 | 31.96 | 0.15 | 61.56 | -10.79 | 0.0228 | 0.0171 |
| JPEG2000 | 1000× | 0.46 | 34.15 | 0.12 | 86.10 | -20.63 | 0.0231 | 0.0433 |
| VAPOR | 64× | 0.81 | 30.45 | 0.14 | 64.96 | -21.72 | 0.0231 | 0.0109 |
| NF | 1167× | **0.30** | **36.87** | **0.07** | 54.04 | 0.78 | 0.0199 | 0.0181 |
| PINC-NF | 1167× | 0.34 | 35.43 | 0.09 | **2.46** | **13.07** | **0.0062** | 0.0161 |
| AE + EVA | 716× | 0.40 | 35.55 | 0.11 | 11.74 | 6.79 | 0.0176 | 0.0104 |
| VQ-VAE + EVA | **77368×** | 0.49 | 32.62 | 0.14 | 30.55 | 7.65 | 0.0166 | **0.0100** |

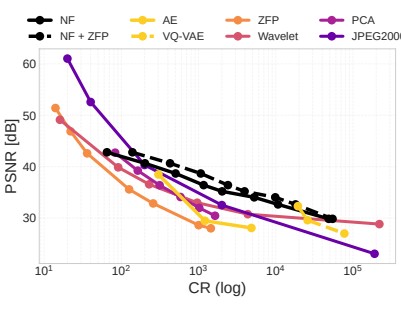

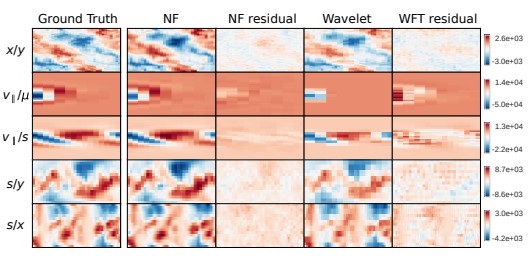

(a) Performance rate scaling.

(b) $\boldsymbol{f}$ along different axes.

Figure 2: **Left:** Compression performance rate-distortion as Peak Signal to Noise Ratio (PSNR) on Compression Rate (CR) on 3 randomly sampled timesteps from 10 trajectories (30 total samples). **Right:** qualitative visualization as 2D projection of sampled 5D densities $\boldsymbol{f}$ with residuals.

Table 1 quantitatively summarizes the results of our analysis. At equivalent compression rate (CR), neural fields and autoencoders improve on traditional methods on compression, as well as integrated quantities and turbulence metrics. However, especially integral metrics exhibit discrepancies from the ground-truth. This motivates the need for PINC which imposes a soft-constraint on the optimization procedure to preserve such quantities. This is verified by comparing NF to PINC-NF, which reveals great improvements on integral errors at a modest reconstruction degradation. Furthermore, WD decreases by an order of magnitude for $\overline{k_y^{\text{spec}}}$. Interestingly, we do not observe an improvement on $\overline{Q^{\text{spec}}}$, possibly due to competing objectives. Qualitative examples of reconstructions for $\boldsymbol{f}$ and $\phi$ are in Figure 2b and Figure 3, and extra projections are in Appendix at Figure 12 and 13.

**Performance-rate scaling.** To assess how reconstruction quality scales across compression levels, we train a series of neural fields and autoencoders with progressively larger parameter counts and latent sizes. Training neural fields remains relatively inexpensive, whereas autoencoders become unfeasible in terms of both GPU memory and runtime at lower CRs. Consequently, we train only six autoencoders in total (three standard and three VQ-VAEs), all at comparatively high CRs (>1,000×). Findings reported in Figure 2 suggest that both learned methods present a specific "window" of CRs in which they significantly outperform traditional baselines (namely in the $500 - 10,000\times$ range). Moreover, neural fields also exhibit a favorable exponential decay (linearly in semilog-x), as opposed to super-exponential of others (polynomial in semilog-x). This is supported by neural field compression on other modalities (Dupont et al., 2022b; Bauer et al., 2023). In terms of reconstruction quality, at lower rates ($< 200\times$) neural compression cannot reliably match wavelets or JPEG2000, and at extreme CRs ($> 40,000\times$) they are comparable.

### 4.2 PHYSICS AND TURBULENCE PRESERVATION

**Physical losses ablations.** We verify the impact of each loss term described in Equation (8) by training different models on each term in Section 3.1 and Section 3.3 separately, for both autoencoders and neural fields. Table 4a collects the ablation findings. Training $\mathcal{L}_{\text{int}}$ and $\mathcal{L}_{\text{diag}}$ have similar effects, both improve the integral as well as the diagnostics, with the integral being more informative. The model still gets valuable information on $Q$ and $\phi$ from the gradients through $\mathcal{L}_{\text{diag}}$. In contrast, $\mathcal{L}_{\text{grad}}$ alone has a destabilizing effect, and is only effective when combined with other losses as it is dependent on how accurately the diagnostics (and integrals) are captured. Finally, the composite $\mathcal{L}_{\text{PINC}} = \mathcal{L}_{\text{int}} + \mathcal{L}_{\text{diag}} + \mathcal{L}_{\text{grad}}$ gathers benefits of each component.

Overall both classes of methods greatly improve performance on physical losses when trained on $\mathcal{L}_{\text{PINC}}$, while slightly decreasing $\boldsymbol{f}$ PSNR. The degradation in reconstruction observed for neural fields is connected to the interpretation of the physical loss scaling behaviors (Section 4.2): as minimizing $\mathcal{L}_{\text{PINC}}$ shifts the modes to ones relevant for integrals and diagnostics, some of the dominant ones of $\boldsymbol{f}$ become less represented and the decoded quality slightly degrades. While neural field training is generally consistent, for autoencoders severe instabilities emerge when training jointly

| Model | Loss | $f$ | $\mathcal{L}_Q$ | $\mathcal{L}_\phi$ | $\mathcal{L}_{k_y^{\text{spec}}}$ | $\mathcal{L}_{Q^{\text{spec}}}$ |
|---|---|---|---|---|---|---|
| NF | $\mathcal{L}_{\text{recon}}$ | **38.89** | 48.59 | 4.45 | 3.71 | 1.52 |
| | $+\mathcal{L}_{\text{int}}$ | 36.68 | **10.35** | 2.55 | 1.61 | 1.42 |
| | $+\mathcal{L}_{\text{diag}}$ | 38.76 | 41.39 | 2.25 | 1.67 | **1.32** |
| | $+\mathcal{L}_{\text{grad}}$ | 37.29 | 63.94 | 44.18 | * | 2.0 |
| | $+\mathcal{L}_{\text{PINC}}$ | 38.28 | 28.03 | **1.41** | **0.24** | 1.41 |
| VQ-VAE | $\mathcal{L}_{\text{recon}}$ | 26.96 | 86.21 | * | * | 91.68 |
| | $+\mathcal{L}_{\text{PINC}}$ | 27.73 | 85.06 | 103.50 | * | * |
| + EVA | $+\mathcal{L}_{\text{PINC}}$ | **32.21** | **27.73** | **40.81** | **284.96** | **59.84** |

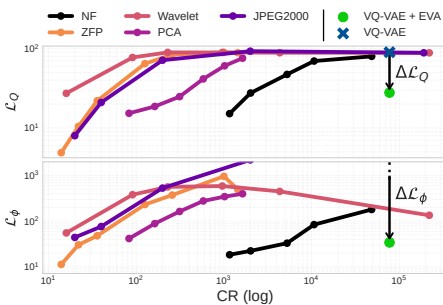

(a) PINC losses ablation table.

(b) Physics performance rate scaling.

Figure 4: **Left:** ablations of the PINC losses (colored blocks) from Equation (8) for neural fields and autoencoders. Both on 3 randomly sampled timesteps from 10 trajectories (30 total samples). PSNR reported for $f$. * means $> 100\times$ larger than column average. Bold numbers are per model class. **Right:** Physical losses scaling as $\mathcal{L}_Q$ (top) and $\mathcal{L}_\phi$ (bottom) on Compression Rate (log-log). $\Delta\mathcal{L}$ PINC improvement for VQ-VAE + EVA is reported with the downward arrow.

on $\mathcal{L}_{\text{recon}} + \mathcal{L}_{\text{PINC}}$. Our EVA finetuning procedure is consistently outperforming and more stable than directly training on $\mathcal{L}_{\text{PINC}}$ (bottom of Table 4a).

**Physical scaling.** Similarly to Figure 2a for rate-distortion for the distribution function $f$, Figure 4b shows scaling for heat flux $Q$ and electrostatic potential $\phi$ integral losses as CR is changed. Figure 3 shows projections of the 3D $\phi$ integral and residuals (CR $=\sim$ 1,000$\times$). Traditional compression struggles to capture $\phi$ even at low CR, while models trained on Equation (8) as well as the reconstruction loss (Equation (5)) yield reasonable reconstruction. A possible interpretation is that, since modeling capacity is constrained by high compression, the available "entropy" gets allocated across modes, according to the encoding algorithm. In neural networks, the spectral bias (Rahaman et al., 2019) of MSE training (Equation (5)) implies that high-energy components have priority during training, while lower-energy modes converge slower. PINC

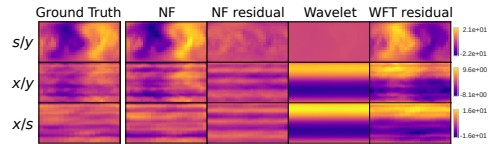

Figure 3: $\phi$ 3D projections.

appears to redistribute some of the energy to more physically relevant modes. For example, the heat flux integral masks low frequencies and rescales high frequencies, giving them more importance.

**Recovering turbulence.** Figure 5 qualitatively shows how well different models capture the direct energy cascade phenomena across different simulations (energy shifting to lower modes over time), by visualizing the *per-timestep* spectras $k_y^{\text{spec}}$ and $Q^{\text{spec}}$ in a log-log plot. The Figure provides a qualitative comparison of turbulence recovery on the temporal axis, in contrast to the steady-state statistics reported in Table 1. **[kwUE, QH9G]** The time snapshots examined in Figure 5 (sampled between $[8.4, 24.4]R/V_r$ with a step size of $\Delta = 2.0R/V_r$) are sampled in the transitional phase where turbulence grows, at the so called *overshoot*. These timesteps are different to those in Table 1. On $k_y^{\text{spec}}$, traditional compression methods already achieve reasonable performance in most cases, but on $Q^{\text{spec}}$ they produce severely nonphysical results (flat curves, negative numbers). Another observation is that, even though non-ML methods have fairly low Wasserstein distance in Table 1, this is not reflected at the overshoot. In contrast, neural fields and VQ-VAE can reproduce the overall profiles consistently, with VQ-VAE excelling at the flux spectra. However, both often fail to capture the higher-frequency magnitudes. The behaviors can be attributed to the spectral bias of neural networks (Rahaman et al., 2019; Teney et al., 2025), where low-frequency (high-energy) components are favored over high-frequencies. Appendix C.7 shows additional cascade plots for all methods and trajectories. **[kwUE, QH9G]** Figure 5b shows that neural compression can significantly outperform traditional methods both on the accumulated energy cascade errors (Equation (3)), as well as the endpoint error of the density function $f$ optical flows (Equation (4)). Note that the EPE reported here differs from the one in Table 1 in that it is applied to the transitional phase instead

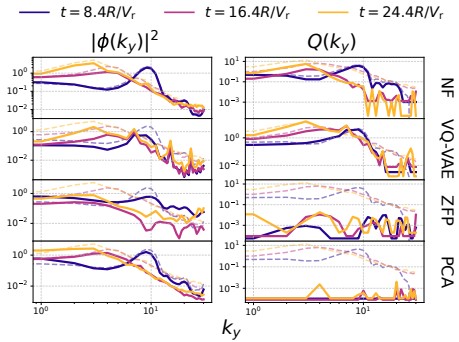

| Model | EPE $\downarrow$ | $EC_{k_y} \downarrow$ | $EC_Q \downarrow$ |
|---|---|---|---|
| ZFP | $0.058_{\pm 0.03}$ | $0.031_{\pm 0.01}$ | $0.715_{\pm 1.30}$ |
| Wavelet | $0.033_{\pm 0.01}$ | $0.031_{\pm 0.01}$ | $0.061_{\pm 0.09}$ |
| PCA | $0.032_{\pm 0.02}$ | $0.032_{\pm 0.01}$ | $0.065_{\pm 0.07}$ |
| JPEG2000 | $\underline{0.027}_{\pm 0.01}$ | $0.032_{\pm 0.01}$ | $0.176_{\pm 0.21}$ |
| NF | $\mathbf{0.017}_{\pm 0.01}$ | $0.030_{\pm 0.01}$ | $0.029_{\pm 0.02}$ |
| PINC-NF | $0.030_{\pm 0.02}$ | $\mathbf{0.011}_{\pm 0.01}$ | $0.015_{\pm 0.00}$ |
| PINC-AE | $0.030_{\pm 0.02}$ | $0.028_{\pm 0.01}$ | $\mathbf{0.005}_{\pm 0.00}$ |
| PINC-VQ-VAE | $0.036_{\pm 0.02}$ | $\underline{0.018}_{\pm 0.01}$ | $\underline{0.008}_{\pm 0.00}$ |

(a) Energy cascade in $k_y^{\text{spec}}$ and $Q^{\text{spec}}$.  (b) **[kwUE, QH9G]** Temporal consistency metrics.

Figure 5: **[kwUE, QH9G] Left:** Energy cascade visualized as the transfer from higher to lower modes as turbulence develops. Plots in loglog scale. **Right:** Quantitative temporal consistency on optical flow endpoint error (EPE) and energy cascade optimal transport (EC). Evaluation on 270 total $\boldsymbol{f}_t^c$s (30 trajectories, 9 timesteps), sampled in the transitional phase where mode growth happens.

of the saturated, statistically steady one. Its purpose is to determine how well the energy cascade and mode growth is reconstructed.

### 4.3 REPRESENTATION SPACE EXPERIMENTS

**Hybrid compression.** Neural methods can further improve the compression rate if coupled with traditional techniques applied in weight space. Similarly to how data can be compressed into a low dimensional representation, network weights are redundant and also lie on a lower-dimensional manifold. This is related to pruning (LeCun et al., 1990; Han et al., 2015), network compression (Hershcovitch et al., 2024), and the lottery ticket hypothesis (Frankle & Carbin, 2019).

Improved compression can be achieved either with (lossless) entropy coding (Hershcovitch et al., 2024) or (lossy) quantization methods (Lindstrom, 2014). We apply both to neural fields and present findings in Table 2. ZipNN is lossless and does not induce any change in performance, while providing a modest improvement in CR. ZFP is lossy with a tolerance of $10^{-3}$, leading to minor performance degradation and a $2.1\times$ improved CR. Both

Table 2: Hybrid compression.

| Metric | | ZFP | ZipNN |
|---|---|---|---|
| Extra CR | | $2.1\times$ | $1.2\times$ |
| $\Delta$ PSNR ($\boldsymbol{f}$) | $\uparrow$ | +2e-4% | 0% |
| $\Delta$ L1 ($Q$) | $\downarrow$ | +8e-3% | 0% |
| $\Delta$ L1 ($\phi$) | $\downarrow$ | +9.5% | 0% |

results are averaged on 60 random samples from 10 trajectories. We also show NF + ZFP in Figure 2a. It closely follows the slope of NF, but is shifted to the right, achieving better CR. Notably, at the higher regimes they appear to converge, suggesting diminishing returns. **[WgAS]** As an utmost example, one can include entropy coding on the VQ-VAE indices, bringing the compression to $121492\times$ (see Appendix C.5).

**Latent (and weight space) interpolation.** Representational consistency and compactness over different snapshots is a desired property of compression methods. It enables temporal coarsening (Ohana et al., 2024; Toshev et al., 2023) by interpolation in weight/latent space resulting in additional gains in CR as not every single snapshot needs to be compressed. To this end, we design an experiment to assess whether PINC models exhibit representational consistency across time. We encode two *extremes* $\boldsymbol{f}_a, \boldsymbol{f}_b$ separated by $\Delta T$ and reconstruct intermediates $\boldsymbol{f}_t$ for $t = a, a + \mathrm{d}T, \dots, b$ by linearly interpolating the representations (latents or weights) $Z_{\boldsymbol{f}_a}$ and $Z_{\boldsymbol{f}_b}$.

For standard autoencoders, latent-space interpolation is a common practice (Berthelot* et al., 2019). In the case of VQ-VAEs, the latents are interpolated before quantization to produce more accurate reconstructions. It is not as straightforward for neural fields, as the parameters are not necessarily canonically ordered and exhibit various neuron symmetries (Hecht-Nielsen, 1990; Godfrey et al., 2022). To address this, we use a *meta neural field* trained on all extremes before finetuning it on each of them separately, ensuring shared initialization and improving alignment. This is similar to the initialization strategy used by Luigi et al. (2023) and Erkoç et al. (2023) to generate an aligned dataset of neural fields.

| Model | PSNR | L1 |
|---|---|---|
| Extremes | 16.7 | 0.87 |
| $f$ (data) | 19.6 | 0.73 |
| NF (weights) | 18.9 | 0.76 |
| AE (latents) | 18.5 | 0.99 |
| VQ-VAE (latents) | 20.5 | 0.69 |

(a) Interpolation inputs.

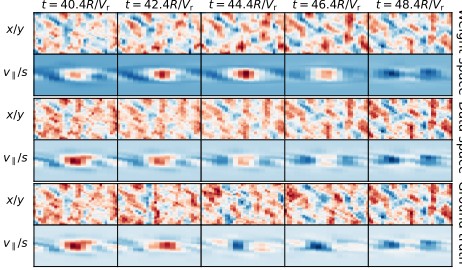

(b) Interpolated slice visualization over time.

Figure 6: **Left:** time coarsening on the middle snapshot $t_m = t_l + \frac{\Delta T}{2}$, showing that representation interpolation outperforms using extremes and is comparable to data space. Results are averaged on 50 (unseen) midpoints on 10 trajectories, with $\Delta T = 8R/V_r$. **Right:** Qualitative visualization of 5D $f$ slices interpolated over time, between the two extremes at $t = 40.4R/V_r$ and $t = 48.4R/V_r$.

Figure 6a provides compelling evidence that linearly interpolating in representation space improves over simply taking the extremes, and approximates linear interpolation in data space. Figure 6b illustrates intermediate reconstructions over time as progressive interpolation between $Z_{f_a}$ and $Z_{f_b}$. However, because the underlying simulations are highly nonlinear accurate linear interpolation is unlikely, hence the low reported PSNR. Regardless, we reckon that these results shows that learned representations are compact and self-consistent over time.

## 5 CONCLUSIONS

Our study provides compelling evidence that Physics-Informed Neural Compression (PINC) improves compression rate while maintaining underlying characteristics for gyrokinetic simulations of plasma turbulence. This is achieved by constraining training to maintain integral quantities and spectral shapes across key dimensions of the 5D fields. We anticipate that this approach can potentially be extended to other scientific domains, enabling practitioners to store compressed simulations that accurately capture specified physical phenomena across time and space, something previously infeasible due to storage requirements. These tools could considerably improve data accessibility and transfer, accelerating research across scientific communities.

Our work paves the way for fruitful future avenues. The compression methods presented in this work could be combined with *neural operators*, nonlinearly evolving them in time. A major benefit of this is a significant reduction in dataset size required to train a surrogate model. Orthogonally, exploring physics inspired "functsets" (Dupont et al., 2022a; Jo et al., 2025) could be a valuable direction to further improve compression of neural fields for transient simulations and enable in-transit processing of data. Related approaches in this regard include continual learning (Yan et al., 2021; Woo et al., 2025), and in general ways to incorporate temporal dynamics into the training to enable on-the-fly (*in-situ*) compression of simulation snapshots.

**Limitations.** First, we do not incorporate temporal information during PINC training, which we expect to especially improve on temporal consistency. Due to the computational complexity of training neural fields and especially autoencoders, this avenue is left to future work. Second, the computational requirements are substantial, mirrored in the training times (Table 10). Even for neural fields, compression times are rather high and a modest GPU is required. Finally, the proposed physics-informed losses are specific to gyrokinetics, limiting transferability to other scientific areas beyond plasma physics. **[Dmk1, kwUE, QH9G]** Concurrent neural compression works, such as Momenifar et al. (2022) for fluid dynamics and Cranganore et al. (2025) for General Relativity, are also problem-specific. To our knowledge there is no general loss reformulation that is applicable to any problem, and we reserve extending PINC to other domains as future work. We postulate that the strategies and methodologies used for gyrokinetics-PINC, for example the stabilization with EVA finetuning used for the autoencoders, can be successfully extended to other sources. Moreover, with the right adjustments on the physics, the evaluation pipeline is applicable to any spatiotemporal system where compression over time is not possible or exceedingly costly.

## REPRODUCIBILITY STATEMENT

Training and experiment code is submitted as a zip file in the supplementary materials. It contains autoencoders, neural fields and baseline implementation, as well as the configuration files used to obtain the paper results. The readme briefly outlines the code structure and describes how to start autoencoder/neural field training runs. Some further information on training is already present in the Method and Results sections, as well as dedicated sections in the Appendix. Unfortunately, the dataset is not easily distributable due to its size. It was generated with the GKW (Peeters et al., 2009) flux tube gyrokinetic numerical solver, as detailed in Appendix B. A template for the configuration file used by GKW to start a run is included in the supplementary materials (`data_generation/` directory). Parameter ranges used to generate the dataset are included both in the supplementary as well as in Appendix B for transparency. **[QH9G]** We release a validation dataset along with neural field weights and autoencoder checkpoints at this link.

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

## LLM USAGE DISCLOSURE

In general, LLM tools were used to refine writing in multiple parts of the paper, such as introduction and experiment section. This very paragraph is written by a human, polished by GPT-5. Some of the literature cited in the related work and introduction sections was also fetched by GPT-5. DeepSeek-R1 and GPT-5 were additionally make visualizations prettier, speed up the development of plotting functions, and dump results neatly into tables. Beyond that, they were not used to a significant degree in other parts of the code, as neither Copilot nor Cursor are used by the main author. AI assistants were strictly editors and decorators – they were **not** involved in ideation, reordering ideas, or at any higher or lower conceptual level. Rebuttal update: VAPOR was re-implemented with heavy usage of Gemini 2.5 (pro).

## A   GYROKINETICS

Gyrokinetics (Frieman & Chen, 1982; Krommes, 2012; Peeters et al., 2009) is a reduced form of Plasma kinetics that is computationally more efficient and can be use to locally simulate Plasma behavior within a so-called *flux tube* in the torus. Local gyrokinetics is a theoretical framework to study plasma behavior on perpendicular spatial scales comparable to the gyroradius, i.e., the radius of circular motion exhibited by charged particles in a magnetic field, and frequencies much lower than the particle cyclotron frequencies, i.e., the frequency at which charged particles spiral around magnetic field lines due to the Lorentz force. Gyrokinetics models the time evolution of electrons and ions via the distribution function $\boldsymbol{f}$, which is based on 3D coordinates, their parallel and perpendicular velocities, together with the angle w.r.t. the field lines. However, the latter dimension is averaged out by modelling only the so-called guiding center of a particle instead of its gyral movement. Furthermore, instead of modelling the perpendicular velocity, usually only its magnitude is considered, which is also referred to as the magnetic moment $\mu$. Hence, the 5D gyrokinetic distribution function can be written as $\boldsymbol{f} = \boldsymbol{f}(k_x, k_y, s, v_\parallel, \mu)$, where $k_x$ and $k_y$ are spectral coordinates, $s$ is the toroidal coordinate along the field line, and $v_\parallel$ the parallel velocity component. The perturbed time-evolution of $\boldsymbol{f}$, for each species (ions and electrons), is governed by

$$\underbrace{\frac{\partial \boldsymbol{f}}{\partial t} + (v_\parallel \boldsymbol{b} + \boldsymbol{v}_D) \cdot \nabla \boldsymbol{f} - \frac{\mu B}{m} \frac{\boldsymbol{B} \cdot \nabla B}{B^2} \frac{\partial \boldsymbol{f}}{\partial v_\parallel}}_{\text{Linear}} + \underbrace{\boldsymbol{v}_\chi \cdot \nabla \boldsymbol{f}}_{\text{Nonlinear}} = S \, , \tag{9}$$

where $v_\parallel \boldsymbol{b}$ is the motion along magnetic field lines, $\boldsymbol{b} = \boldsymbol{B}/B$ is the unit vector along the magnetic field $\boldsymbol{B}$ with magnitude $B$[1], $\boldsymbol{v}_D$ the magnetic drift due to gradients and curvature in $\boldsymbol{B}$, and $\boldsymbol{v}_\chi$ describes drifts arising from the $\boldsymbol{E} \times \boldsymbol{B}$ force, a key driver of plasma dynamics. Finally, S is the source term that represents the external supply of energy. The term $\boldsymbol{v}_\chi \cdot \nabla \boldsymbol{f}$ models the nonlinear interaction between the distribution function $\boldsymbol{f}$ and its velocity space integral $\phi$, and it describes turbulent advection. The resulting nonlinear coupling constitutes the computationally most expensive term.

### A.1   DERIVATION OF THE GYROKINETIC EQUATION

We begin with the Vlasov equation for the distribution function $\boldsymbol{f}(\boldsymbol{r}, \boldsymbol{v}, t)$:

$$\frac{\partial \boldsymbol{f}}{\partial t} + \mathbf{v} \cdot \nabla \boldsymbol{f} + \frac{q}{m} \left( \boldsymbol{E} + \boldsymbol{v} \times \boldsymbol{B} \right) \cdot \nabla_v \boldsymbol{f} = 0 \tag{10}$$

The Vlasov equation describes the conservation of particles in phase space in the absence of collisions. Here, $\boldsymbol{r} = (x, y, z)$ and $\boldsymbol{v} = (v_x, v_y, v_z)$ correspond to coordinates in the spatial and the velocity domain, respectively. Hence the Vlasov equation is a 7D (including time) PDE representing the density of particles in phase space at position $\boldsymbol{r}$, velocity $\boldsymbol{v}$, and time. The term $\nabla_{\boldsymbol{v}} \boldsymbol{f}$ describes the response of the distribution function to accelerations of particles and $\frac{q}{m} (\mathbf{E} + \mathbf{v} \times \mathbf{B})$ denotes the Lorentz force, which depends on particle charge $q$ and mass $m$, as well as electric field $\boldsymbol{E}$ and magnetic field $\boldsymbol{B}$. Finally, the advection (or convection) term $\boldsymbol{v}\nabla \boldsymbol{f}$ describes transport of the distribution functon through space due to velocities.

---

[1]We adopt uppercase notation for vector fields $\boldsymbol{E}$ and $\boldsymbol{B}$ to adhere with literature.

To derive the *gyrokinetic equation*, we transform from particle coordinates to guiding center coordinates $(\mathbf{R}, v_\parallel, \mu, \theta)$, where $\mu = \frac{mv_\perp^2}{2B}$ is the magnetic moment, $\theta$ the gyrophase, which describes the position of a particle around its guiding center as it gyrates along a field line, and $\boldsymbol{R}$ is the coordinate of the guiding center.

Assuming the time scale $L$ at which the background field changes is much longer than the gyroperiod with a small Larmor radius $\rho \ll L$, we can *gyroaverage* to remove the dependency on the gyrophase $\theta$, yielding:

$$\frac{\partial \boldsymbol{f}}{\partial t} + \dot{\boldsymbol{R}} \cdot \nabla \boldsymbol{f} + \dot{v}_\parallel \frac{\partial \boldsymbol{f}}{\partial v_\parallel} = 0 \tag{11}$$

### A.1.1 LINEAR TERMS

The unperturbed (background) motion of the guiding center is governed by:

$$\dot{\boldsymbol{R}} = v_\parallel \boldsymbol{b} + \boldsymbol{v}_D \tag{12}$$

$$\dot{v}_\parallel = -\frac{\mu}{m} \boldsymbol{b} \cdot \nabla \boldsymbol{B} \tag{13}$$

Here, $\mathbf{b} = \boldsymbol{B}/B$ is the unit vector along the magnetic field, and $\boldsymbol{v}_D$ represents magnetic drifts. Substituting into the kinetic equation yields

$$\frac{\partial \boldsymbol{f}}{\partial t} + (v_\parallel \boldsymbol{b} + \boldsymbol{v}_D) \cdot \nabla \boldsymbol{f} - \frac{\mu}{m} \boldsymbol{b} \cdot \nabla \boldsymbol{B} \frac{\partial \boldsymbol{f}}{\partial v_\parallel} = 0 \tag{14}$$

We can express the magnetic gradient term using:

$$\boldsymbol{b} \cdot \nabla \boldsymbol{B} = \frac{\boldsymbol{B} \cdot \nabla \boldsymbol{B}}{B} \tag{15}$$

so that:

$$\frac{\mu}{m} \boldsymbol{b} \cdot \nabla \boldsymbol{B} = \frac{\mu B}{m} \frac{\boldsymbol{B} \cdot \nabla \boldsymbol{B}}{\boldsymbol{B}^2} \tag{16}$$

### A.1.2 NONLINEAR TERM

Fluctuating electromagnetic potentials $\delta\phi, \delta\boldsymbol{A}$ induce E×B and magnetic flutter drifts. We define the gyroaveraged generalized potential as

$$\chi = \langle \boldsymbol{\phi} - \frac{v_\parallel}{c} A_\parallel \rangle, \tag{17}$$

where $\boldsymbol{A}_\parallel$ is the parallel component of the vector potential, $\langle \cdot \rangle$ denotes the gyroaverage, and $c$ is the speed of light, which is added to ensure correct units. $\phi$ is the electrostatic potential, the computation of which involves an integral of $\boldsymbol{f}$ over the velocity space (see eq. 1.41 in the GKW manual [2] for a complete description).

This gives rise to the drift

$$\mathbf{v}_\chi = \frac{c}{B} \mathbf{b} \times \nabla\chi, \tag{18}$$

and yields the nonlinear advection term $\mathbf{v}_\chi \cdot \nabla \boldsymbol{f}$.

### A.1.3 FINAL EQUATION

We arrive at the gyrokinetic equation in split form:

$$\frac{\partial \boldsymbol{f}}{\partial t} + (v_\parallel \mathbf{b} + \mathbf{v}_D) \cdot \nabla \boldsymbol{f} - \frac{\mu B}{m} \frac{\boldsymbol{B} \cdot \nabla \boldsymbol{B}}{\boldsymbol{B}^2} \frac{\partial \boldsymbol{f}}{\partial v_\parallel} + \mathbf{v}_\chi \cdot \nabla \boldsymbol{f} = S \tag{19}$$

---

[2]https://bitbucket.org/gkw/gkw/src/develop/doc/manual/

Here, $S$ represents external sources, collisions, or other drive terms. To enhance the tractability of Equation (9), the distribution function $\boldsymbol{f}$ is usually split into equilibrium and perturbation terms

$$\boldsymbol{f} = \boldsymbol{f}_0 + \delta\boldsymbol{f} = \underbrace{\boldsymbol{f}_0 - \frac{Z\phi}{T}\boldsymbol{f}_0}_{\text{Adiabatic}} + \underbrace{\frac{\partial h}{\partial t}}_{\text{Kinetic}}, \tag{20}$$

where $\boldsymbol{f}_0$ is a background or equilibrium distribution function, $T$ the particle temperature, $Z$ the particle charge, $\phi$ the electrostatic potential, and $\delta f$ the total perturbation to the distribution function, which comprises of *adiabatic* and *kinetic* response. The adiabatic term describes rapid and passive responses to the electrostatic potential that do not contribute to turbulent transport, while the kinetic term governs irreversible dynamics that facilitate turbulence. Numerical codes, such as GKW (Peeters et al., 2009), rely on solving for $\delta f$ instead of $\boldsymbol{f}$. A common simplification is to assume that electrons are adiabatic, which allows us to neglect the kinetic term in the respective $\delta\boldsymbol{f}$. Hence, the respective $\boldsymbol{f}$ for electrons ($\boldsymbol{f}_e$) does not need to be modelled, effectively halving the computational cost.

## B DATASET

The simulations used for both the autoencoder training (26 trajectories) and the evaluation (10 trajectories) are generated with the numerical code GKW (Peeters et al., 2009). They are sampled by varying four parameters: $R/L_t$, $R/L_n$, $\hat{s}$, and $q$, which significantly affect emerging turbulence in the Plasma.

- $R/L_t$ is the ion temperature gradient, which is the main driver of turbulence.
- $R/L_n$ is the density gradient, whose effect is less pronounced. It can have a stabilizing effect, but can sometimes also lead to increased turbulence.
- $\hat{s}$ denotes magnetic shearing, hence it usually has a stabilizing effect as more magnetic shearing results in better isolation of the Plasma.
- $q$ denotes the so-called safety factor, which is the inverse of the rotational transform and describes how often a particle takes a poloidal turn before taking a toroidal turn.

We specify the ranges for sampling the four parameters as $R/L_T \in [1, 12]$, $R/L_n \in [1, 7]$, $q \in [1, 9]$, and $\hat{s} \in [0.5, 5]$. Additionally, we also vary the noise amplitude of the initial condition (within $[1e-5, 1e-3]$).

To make storage more feasible, simulations are time-coarsened by saving snapshot every 60. Each GKW run with the specified configurations takes around $\sim$6 hours (76 cores, Intel Ice Lake 4.1GHz CPU) and $\sim$60GBs of storage.

## C IMPLEMENTATION DETAILS

### C.1 METRICS

We evaluate reconstruction with spatial and physical metrics. Since gyrokinetic data is complex-valued, we can also apply complex-generalizations of common metrics.

**Complex L1 Loss.** Given two complex-valued fields $z_1, z_2 \in \mathbb{C}^N$, the complex L1 loss is:

$$\text{cL1}(z_1, z_2) = \langle |\Re(z_1 - z_2)| + |\Im(z_1 - z_2)| \rangle = \langle |z_1 - z_2|_1 \rangle$$

where $\langle \cdot \rangle$ denotes the average over all dimensions and $|\cdot|_1$ is the L1 norm of the complex difference.

**Wasserstein Distance.** The Wasserstein distance measures the minimum cost of transforming one probability distribution into another, where the cost is proportional to the distance the probability mass must be moved. It provides a meaningful metric to compare distributions even when they have non-overlapping support, making it particular useful in machine learning and optimal transport problems. In our case, we normalize the spectra so that their total sum is one, ensuring they represent comparable probability distributions.

The Wasserstein distance between two probability distributions $P$ and $Q$ is defined as:

$$W_p(P, Q) = \left( \inf_{\gamma \in \Gamma(P,Q)} \int \|x - y\|^p \, d\gamma(x, y) \right)^{\frac{1}{p}}$$

**Peak Signal-to-Noise Ratio.** Peak signal-to-noise ratio (PSNR) quantifies the ratio between the maximum possible power of a signal and the power of noise corrupting its representation, typically expressed in decibels (dB) due to the wide dynamic range of signals.

$$\text{PSNR}(x_1, x_2) = 10 \cdot \log_{10} \left( \frac{\max(x_1)^2}{\text{MSE}(x_1, x_2)} \right)$$

where $\text{MSE}(x_1, x_2)$ is the mean squared error between the real-valued fields $x_1$ and $x_2$.

The PSNR for complex-valued fields we defined as:

$$\text{cPSNR}(z_1, z_2) = 10 \cdot \log_{10} \left( \frac{\max(|z_1|)^2}{\text{cMSE}(z_1, z_2)} \right)$$

**Bits Per Pixel (BPP).** The BPP measures compression efficiency. Given a discrete representation of a field $z$ and its compressed encoding, the bits per pixel is defined as

$$\text{BPP} = \frac{\text{Total number of bits used to encode } z}{\text{Number of spatial points in } z}.$$

Lower BPP values indicate higher compression, while higher BPP generally corresponds to more faithful reconstruction.

## C.2   TEMPORAL METRICS

**Optical Flow and End-Point Error (EPE).** Optical flow estimates the apparent motion between consecutive frames of a sequence by computing spatial gradients and temporal derivatives, here implemented using a simplified Horn–Schunck finite differencing method (Horn & Schunck, 1981). Since optical flow is typically implemented in 2D + time, we rearrange each $f$ into a 2D array as $(v_\parallel \cdot \mu) \times (s \cdot y \cdot x)$ For two consecutive frames $x(t)$ and $x(t+1)$, the averaged spatial gradients

$$x_x = \frac{1}{2} \left( \partial_x x(t) + \partial_x x(t+1) \right), \quad x_y = \frac{1}{2} \left( \partial_y x(t) + \partial_y x(t+1) \right).$$

The optical flow field $\mathbf{F}$, representing the motion gradient between frames,

$$\mathbf{F} = \left[ -\frac{x_x \cdot (x(t+1) - x(t))}{x_x^2 + x_y^2}, \; -\frac{x_y \cdot (x(t+1) - x(t))}{x_x^2 + x_y^2} \right],$$

Given two sequences of $N$ frames $x_1$ and $x_2$, the EndPoint Error (EPE) (Baker et al., 2011) is the mean squared difference of the flow vectors $\mathbf{F}_1^{(i)}$ and $\mathbf{F}_2^{(i)}$ over time.

$$\text{EPE}(x_1, x_2) = \frac{1}{N} \sum_{i=1}^{N} \|\mathbf{F}_1^{(i)} - \mathbf{F}_2^{(i)}\|_2^2,$$

where $N$ is the total number of spatial points across all frames.

## C.3   TRADITIONAL COMPRESSION

In the following paragraphs we briefly describe how the traditional compressions were implemented.

**ZFP Compression.** ZFP Lindstrom (2014) is a compression library for numerical arrays designed for fast random access. It partitions the data into small blocks (typically $4 \times 4 \times 4$ elements for 3D data) and transforms them into a decorrelated representation using an orthogonal block transform.

The transformed coefficients are quantized according to a user-specified tolerance, then entropy-coded to produce a compact bitstream. High-speed random access and both lossy and lossless are possible, making ZFP a very common choice for scientific data storage.

We rearrange $\boldsymbol{f}$ into a 3D array as $((v_{\parallel} \times \mu) \times (s \times y) \times x)$ for ZFP block-based compression scheme (up to 3D), and compress with ZFP with a specified absolute error tolerance. The compressed representation is a compact byte representation. Reconstruction is performed by decompressing with ZFP and reshaping the output back to the original tensor layout.

**Wavelet Compression.** Discrete wavelet transform (DWT) is applied using the level 1 Haar wavelet. The multi-dimensional array is decomposed into wavelets (coefficient and slices). To achieve lossy compression, coefficients are pruned based on a fixed threshold dependent on the desired compression ratio, effectively discarding small high-frequency components. Reconstruction is performed by inverting the DWT.

**Principal Component Analysis Compression.** $\boldsymbol{f}$ is reshaped into a 2D array $((v_{\parallel} \cdot \mu \cdot s) \times (x \cdot y))$, by rearranging together the velocity space $v_{\parallel}, \mu$ with the field line $s$ and the spatial coordinates $x, y$. PCA is applied on the flattened spatial components, retaining a fixed number of principal components dependent on the desired compression ratio ($N = 2$ for $1,000\times$ from Table 1). The compressed representation consists of the principal components, the mean vector, and the explained variance. Reconstruction is achieved by projecting back to the original space, followed by reshaping to the original dimensions.

**JPEG2000 Compression.** $\boldsymbol{f}$ is first reshaped into a 2D image-like representation of shape $((v_{\parallel} \cdot \mu \cdot s) \times (x \cdot y))$, by flattening the velocity space and spatial dimensions. Each channel is independently normalized to the $[0, 1]$ range and quantized to 16-bit unsigned integers. The images are then encoded using the JPEG2000 standard (Christopoulos et al., 2000) at a target quality factor $Q$ that determines the compression ratio. The compressed representation consists of the code-stream size and channelwise normalization statistics (minimum and maximum). Reconstruction is performed by decoding the JPEG2000 bitstream, rescaling back to floating-point values, and unflattening back to the original tensor dimensions.

## C.4 VAPOR

VAPOR (Choi et al., 2021) combines a VQ-VAE van den Oord et al. (2017) compressor and a a Fourier Neural Operator (FNO) (Li et al., 2021) Refiner sequentially. The VQ-VAE provides extreme compression by reducing the size of the original data, and the FNO Refiner then refines the VQ-VAE's coarse output to restore fidelity, achieving both high compression and high accuracy. We utilize a VQ-VAE with Exponential Moving Average (EMA) updates to compress the data $\boldsymbol{f}$. This forms the first stage of the overall architecture. The FNO refiner stage uses a residual structure to efficiently learn and apply the high-frequency corrections needed to match the ground-truth solution, taking the VQ-VAE initial reconstruction as input.

Finally, a core component of Choi et al. (2021) is the specialized physics loss $\mathcal{L}_{\text{physics}}$, employed to enforce conservation laws. This loss computes the MSE between the predicted and ground-truth values of density, momentum, and energy:

$$
\mathcal{L}_{\text{physics}} = \text{MSE}\left( \sum_{v_{\parallel}, v_{\perp}} \boldsymbol{f}_{\text{pred}}, \sum_{v_{\parallel}, v_{\perp}} \boldsymbol{f}_{\text{gt}} \right) + \text{MSE}\left( \sum_{v_{\parallel}, v_{\perp}} \boldsymbol{f}_{\text{pred}}\, v_{\parallel}, \sum_{v_{\parallel}, v_{\perp}} \boldsymbol{f}_{\text{gt}}\, v_{\parallel} \right)
$$

$$
+ \text{MSE}\left( \sum_{v_{\parallel}, v_{\perp}} \boldsymbol{f}_{\text{pred}}\, \tfrac{1}{2} m_s v_{\parallel}^2, \sum_{v_{\parallel}, v_{\perp}} \boldsymbol{f}_{\text{gt}}\, \tfrac{1}{2} m_s v_{\parallel}^2 \right).
$$

This loss is added to the standard reconstruction and VQ losses during training to obtain the final VAPOR loss: $\mathcal{L} = \mathcal{L}_{\text{recon}} + \mathcal{L}_{\text{VQ}} + \mathcal{L}_{\text{physics}}$.

## C.5 AUTOENCODERS

**Architecture and Conditioning.**

The autoencoder and VQ-VAE baselines are built on a 5D Swin Transformer architecture (Galletti et al., 2025; Paischer et al., 2025a), which extends the shifted window attention mechanism to handle high-dimensional scientific data. Figure 7 illustrates the 5D windowed multi-head self-attention (W-MSA) and shifted windowed multi-head self-attention (SW-MSA) layers,

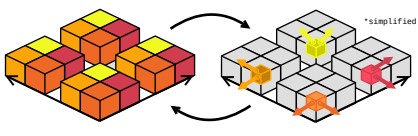

W-MSA (layer l)     SW-MSA (layer l+1)

Figure 7: 5D swin attention.

where blocks of the same color indicate the receptive field of local attention within each window. Our implementation incorporates several stability and performance enhancements: gated attention mechanisms (Qiu et al., 2025) for improved training stability, combined positional encodings using both Relative Positional Bias (Liu et al., 2021) and Rotary Position Embedding (RoPE) (Su et al., 2023) to capture spatial relationships across all five dimensions, and GELU activations (Hendrycks & Gimpel, 2023) throughout the network. Each model uses four Swin blocks with 16 attention heads, followed by a single downsampling level before the bottleneck. All models are conditioned on four key gyrokinetic parameters: the ion temperature gradient ($R/L_t$), density gradient ($R/L_n$), magnetic shear ($\hat{s}$), and safety factor ($q$). Conditioning is implemented via DiT-style modulation (Peebles & Xie, 2023), where conditioning embeddings provide scale, shift, and gating parameters for each transformer layer, enabling physics-aware feature adaptation.

**Data Preprocessing.** The 5D distribution function $[v_\parallel, \mu, s, x, y]$ is represented as complex values with real and imaginary components, initially providing two channels. We apply two key preprocessing steps that affect the channel structure. First, we decomposes each field into zonal flow ($k_y = 0$ mode) and turbulent fluctuation components by computing the mean across the $k_y$ dimension and concatenating the zonal flow and turbulent fluctuation, doubling the channels to four. This separation is essential as zonal flows exhibit fundamentally different physics from turbulent modes. Second, we reshape the magnetic moment dimension $\mu$, into the channel dimension, expanding from four to 32 channels. This allows independent processing of each $\mu$ slice.

**Compression Configurations.** We evaluate multiple compression ratios by varying patch and window sizes. For autoencoders, three configurations achieve compression ratios of 302, 1208, and 2865 using patch sizes $(2, 0, 2, 5, 2)$, $(4, 0, 2, 5, 4)$, and $(6, 0, 3, 5, 6)$ with corresponding window sizes $(8, 0, 4, 9, 8)$, $(4, 0, 4, 9, 4)$, and $(6, 0, 6, 9, 6)$. The zero in the second position corresponds to the $\mu$ dimension, which is not spatially patched due to the decoupling preprocessing step. All variants use latent dimension 1024, and compress in a last linear projection to the bottleneck dimension of 32.

**VQ-VAE Variants.** VQ-VAE uses the same spatial compression configurations but replaces the continuous bottleneck with vector quantization using the implementation from `vector-quantize-pytorch`[3]. The bottleneck projects to 128-dimensional embeddings, which are quantized using a codebook of 8192 vectors (see Table 3 for complete hyperparameters). The codebook uses exponential moving average updates with a decay rate of 0.99 and employs entropy regularization to encourage codebook utilization. This yields much higher compression ratios of 19342, 25789, and 77368 for the three spatial configurations, as quantized codes can be stored as integers (int16 for codebook size of 8192) rather than float32 values.

**Training Strategy.** Training follows a two-stage approach to ensure stability. For all experiments we use Muon optimizer (Jordan et al., 2024) with a cosine scheduler and a minimum learning rate of $4 \times 10^{-6}$, and weight decay of $1 \times 10^{-5}$. *Stage 1* (200 epochs, batchsize=16, lr=$2 \times 10^{-4}$) trains the base autoencoder using only $\mathcal{L}_{\text{recon}}$ (cMSE). *Stage 2.1* (100 epochs, batchsize=16, lr=$2 \times 10^{-4}$) applies Explained Variance Adaptation (EVA) (Paischer et al., 2025b), which injects LoRA (Hu et al., 2022) weights ($r = 64$, $\alpha = 1$, $\rho = 2.0$, $\tau = 0.99$) into MLP layers while freezing the Stage 1 trained backbone. The loss function switches to cL1 for reconstruction ($\mathcal{L}_{\text{recon}}$ weighted by 10.0) and introduces physics-informed losses: integral losses ($\mathcal{L}_Q$, $\mathcal{L}_\phi$) using scale normaliza-

---

[3] https://github.com/lucidrains/vector-quantize-pytorch

Table 3: VQ-VAE vector quantization hyperparameters.

| Parameter | Value |
|---|---|
| Codebook size | 8192 |
| Embedding dimension | 128 |
| Commitment weight | 0.3 |
| Codebook type | Euclidean |
| EMA decay | 0.99 |
| Entropy loss weight | 0.01 |
| Dead code threshold | 2 |

tion (scale is calculated over training dataset statistics), while spectral losses ($\mathcal{L}_{k_y}$, $\mathcal{L}_{Q^{\text{spec}}}$) employ sum-normalization followed by log-space L1 loss. All physics-informed loss terms are weighted equally at 1.0, with the VQ-VAE commitment loss also weighted by a factor of 10.0 to match the reconstruction weight. Critically, monotonicity constraints ($\mathcal{L}_{\text{iso}}$) are disabled. *Stage 2.2* (20 epochs, batchsize=16, lr=$2\times10^{-4}$) continues with identical settings but enables monotonicity losses ($\mathcal{L}_{\text{iso}}(k_y^{\text{spec,pred}})$, $\mathcal{L}_{\text{iso}}(Q^{\text{spec,pred}})$) to enforce physical constraints only after stable physics-informed reconstruction is achieved.

**Training Stabilization.** End-to-end training of autoencoders with physics-informed neural compression (PINC) losses proves highly unstable due to the conflicting optimization objectives and varying loss magnitudes. The physics-informed terms ($\mathcal{L}_Q$, $\mathcal{L}_\phi$, $\mathcal{L}_{k_y}$, $\mathcal{L}_{Q^{\text{spec}}}$) exhibit severe fluctuations during early training when reconstruction quality is poor, causing certain loss components to dominate the overall objective and destabilizing the learning process. This necessitates the staged training approach, where reconstruction capability is first established before introducing physics constraints.

**Multi-objective Optimization Challenges.** We investigated several multi-objective optimization strategies to enable stable end-to-end training. Gradient normalization methods (Chen et al., 2018), while theoretically appealing, proved computationally prohibitive for our large-scale models, consistently causing out-of-memory errors during backpropagation. Conflict-Free Inverse Gradients (ConFIG) Liu et al. (2024) attempts to resolve conflicting optimization objectives by computing gradient directions that minimize conflicts between tasks through least-squares solutions. However, ConFIG relies on computing stable gradient statistics over multiple training steps to determine optimal gradient directions. When physics-informed losses are computed on poorly reconstructed distribution functions, these losses exhibit extreme fluctuations that prevent ConFIG from establishing stable gradient statistics. The method's gradient balancing becomes ineffective when the underlying loss landscape is highly unstable, as the computed conflict-free directions become unreliable due to the volatile nature of the physics-informed terms during early training phases.

**Hyperparameter Search Limitations.** The computational cost of autoencoder training further complicates optimization. Each full training run requires multiple days on high-end GPUs, making systematic hyperparameter search for end-to-end training impractical. The search space includes not only standard hyperparameters (learning rates, batch sizes, architectural choices) but also the relative weighting of several distinct loss components, creating a prohibitively large optimization landscape. This computational constraint reinforces the necessity of our staged approach, which reduces the hyperparameter search to manageable subspaces for each training phase.

**[WgAS] Codebook usage and Entropy Encoding.** The VQ-VAE quantizes the continuous latent space into discrete integer indices ('codes') ranging from 0 to 8191 (codebook size). Each code represents a learned pattern in the distribution function. Standard storage uses fixed-width encoding $\log_2(8192) = 13$ bits per code. However, empirical analysis reveals non-uniform usage: frequent codes dominate (common turbulent structures), while rare codes occur sporadically. This imbalance enables lossless compression via variable-length entropy coding. Our VQ-VAE achieves 71.4% codebook utilization (5846/8192 entries). The sorted codebook frequencies follow Zipf's law, which suggest common flow patterns use frequent codes, while rare events retain dedicated codes.s Further, we measure this redundancy using Shannon entropy $-\sum_i p_i \log_2(p_i)$ where $p_i$ is

the empirical probability of code $i$. Our dataset yields $H \approx 10.5$ bits, indicating that optimal encoding requires only 10.5 bits per code on average, compared to the 13-bit fixed-width baseline. We implement Huffman coding (Huffman, 1952), which constructs a binary tree from code frequencies: frequent codes receive short bit sequences, rare codes longer sequences. The Huffman tree guarantees lossless decoding via bit-by-bit traversal. On our test set, Huffman encoding achieves $1.56\times$ additional compression over fixed-width storage, reducing average code length from 13 to 10.7 bits per code. Combined with VQ-VAE quantazation ($77368\times$), the total pipelines achieves $121492\times$ compression, going from 723.5GB (uncompressed) to 5.96MB (VQ-VAE + Huffman), instead of 9.32MB (VQ-VAE).

## C.6 Neural fields

Neural fields are trained by representing the distribution function as a continuous signal, taking coordinates as inputs. A dataset consists, for a given simulation, of the 5D density function $f$ at a specific timestep, and the 5D grid coordinates of each cell. Data normalization is applied both to the field values and to the coordinates.

An MLP with SiLU activations (Elfwing et al., 2017), 64 hidden dimension, five layers with skip connections and using a discrete hash to map matrix indices to learnable embeddings is optimized using AdamW (Loshchilov & Hutter, 2019), with cosine annealing learning rate scheduling decaying the learning rate from $5e-3$ to $1e-12$ and . Auxiliary optimizers can be used for additional integral losses, also with their scheduler that decays learning rate from $1e-5$ to $1e-12$. The neural field training loop iterates over batches of (2048) coordinates and field values. On a first pass of 20 epochs, the loss $\mathcal{L}_{recon}$ from Equation (5) is fitted. Auxiliary integral losses are trained of such a pretrained model for 100 more epochs, with the whole 5D field as batch.

**ConFIG ablations.** We ablate multi-objective balancing methods such as Conflict-Free Inverse Gradients by Liu et al. (2024) to attempt to stabilize training on the PINC loss terms. Table 4 compares AdamW training (as reported in Table 1) and neural fields complemented with momentum ConFIG with ordered loss selector. Results are similar, with regular AdamW achieving better physical losses and ConFIG being more stable overall.

Table 4: Ablations of NF trained with AdamW and Conflict-Free Inverse Gradients.

| | | Compression $f$ | | | Integrals $Q, \phi$ | | Turbulence $Q^{\text{spec}}, k_y^{\text{spec}}$ | |
| --- | --- | --- | --- | --- | --- | --- | --- | --- |
| | CR | L1 $\downarrow$ | PSNR $\uparrow$ | BBP $\downarrow$ | L1($Q$) $\downarrow$ | PSNR($\phi$) $\downarrow$ | WD($\overline{k_y^{\text{spec}}}$) $\downarrow$ | WD($\overline{Q^{\text{spec}}}$) $\downarrow$ |
| PINC-NF (AdamW) | $1163\times$ | 0.32 | 36.29 | 0.165 | **9.75** | **14.53** | **0.0057** | 0.0170 |
| PINC-NF (SGD+ConFIG) | $1163\times$ | **0.29** | **37.18** | 0.165 | 44.23 | 6.35 | 0.0164 | **0.0163** |

**Neural field ablations.** A broad range of architectures was explored, starting from SIREN (Sitzmann et al., 2020), WIRE Saragadam et al. (2023) and an MLP with different activations (Fukushima, 1969; Hendrycks & Gimpel, 2023; Elfwing et al., 2017). Table 5 summarizes the search space.

Table 5: Neural field search space summary. $w_0$ values are only for SIREN and WIRE architectures.

| Knob | Range |
| --- | --- |
| Activations | Sine, Gabor, ReLU, SiLU, GELU |
| Coordinate embedding | Linear, SinCos, Discrete |
| $w_0^{\text{initial}}$ | 0.1, 0.5, 1.0 |
| $w_0^{\text{hidden}}$ | 0.5, 2.0, 10.0 |
| Skip connections | Yes, No |
| Learning rate | $1e-2, 5e-3$ |

An extensive grid search search was conducted evaluating every combination from Table 5 in the $\sim 1{,}000\times$ compression regime, on 12 randomly sampled density fields $f$ from four different tra-

jectories. For simplicity we use PSNR of $f$ as the selection metric. All models are trained for 10 epochs using the AdamW optimizer Loshchilov & Hutter (2019) with a batch size of 2048. A total of $12 \cdot 36(\text{SIREN}) + 12 \cdot 18(\text{WIRE}) + 12 \cdot 18(\text{MLP}) = 864$ neural fields were trained for this ablation. The results from Tables 6, 7, and 8 suggest that MLP with SiLU activation, skip connections and discrete index embedding is the most performant setup, as well as the fastest and easiest to tune.

Table 6: MLP grid search combinations.

| Activation | Embedding | Skip | Learning rate | $f$ PSNR |
|---|---|---|---|---|
| SiLU | Discrete | Yes | $5e{-}3$ | 40.53 |
| GELU | Discrete | Yes | $5e{-}3$ | 40.12 |
| SiLU | Discrete | No | $5e{-}3$ | 40.11 |
| GELU | Discrete | No | $5e{-}3$ | 39.96 |
| ReLU | Discrete | Yes | $5e{-}3$ | 39.24 |
| ReLU | Discrete | No | $5e{-}3$ | 38.83 |
| GELU | Linear | No | $5e{-}3$ | 37.06 |
| SiLU | SinCos | No | $5e{-}3$ | 36.88 |
| GELU | SinCos | No | $5e{-}3$ | 36.78 |
| GELU | Linear | Yes | $5e{-}3$ | 36.7 |
| SiLU | Linear | No | $5e{-}3$ | 36.47 |
| GELU | SinCos | Yes | $5e{-}3$ | 36.44 |
| SiLU | Linear | Yes | $5e{-}3$ | 36.09 |
| SiLU | SinCos | Yes | $5e{-}3$ | 35.18 |
| ReLU | SinCos | Yes | $5e{-}3$ | 35.1 |
| ReLU | SinCos | No | $5e{-}3$ | 34.68 |
| ReLU | Linear | No | $5e{-}3$ | 34.45 |
| ReLU | Linear | Yes | $5e{-}3$ | 34.4 |

Table 7: SIREN grid search combinations.

| Embedding | $w_0^{\text{initial}}$ | $w_0^{\text{hidden}}$ | Skip | Learning rate | $f$ PSNR |
|---|---|---|---|---|---|
| Discrete | 0.1 | 0.5 | Yes | $5e{-}3$ | 40.48 |
| Discrete | 0.5 | 0.5 | Yes | $5e{-}3$ | 40.34 |
| Discrete | 0.5 | 0.5 | No | $5e{-}3$ | 40.04 |
| Discrete | 0.1 | 0.5 | No | $5e{-}3$ | 39.97 |
| SinCos | 0.5 | 2.0 | Yes | $5e{-}3$ | 38.24 |
| SinCos | 0.1 | 2.0 | Yes | $5e{-}3$ | 38.19 |
| SinCos | 0.5 | 0.5 | No | $5e{-}3$ | 37.22 |
| SinCos | 0.1 | 0.5 | No | $5e{-}3$ | 37.2 |
| SinCos | 0.1 | 0.5 | Yes | $5e{-}3$ | 36.23 |
| SinCos | 0.5 | 0.5 | Yes | $5e{-}3$ | 36.23 |
| SinCos | 0.1 | 2.0 | No | $5e{-}3$ | 32.58 |
| Discrete | 0.1 | 2.0 | No | $5e{-}3$ | 29.41 |
| SinCos | 0.1 | 5.0 | Yes | $5e{-}3$ | 24.16 |
| SinCos | 0.1 | 5.0 | No | $5e{-}3$ | 24.16 |
| Discrete | 0.1 | 5.0 | No | $5e{-}3$ | 24.16 |
| Discrete | 0.1 | 2.0 | Yes | $5e{-}3$ | 24.16 |
| Discrete | 0.5 | 2.0 | Yes | $5e{-}3$ | 24.16 |
| Discrete | 0.1 | 5.0 | Yes | $5e{-}3$ | 24.16 |
| Discrete | 1.0 | 0.5 | Yes | $5e{-}3$ | 10.1 |
| Discrete | 1.0 | 0.5 | No | $5e{-}3$ | 10.03 |
| SinCos | 1.0 | 2.0 | Yes | $5e{-}3$ | 9.57 |
| SinCos | 1.0 | 0.5 | No | $5e{-}3$ | 9.29 |
| SinCos | 1.0 | 0.5 | Yes | $5e{-}3$ | 9.04 |
| SinCos | 1.0 | 2.0 | No | $5e{-}3$ | 8.74 |
| SinCos | 0.5 | 2.0 | No | $5e{-}3$ | 8.43 |
| Discrete | 1.0 | 2.0 | No | $5e{-}3$ | 6.99 |
| Discrete | 0.5 | 2.0 | No | $5e{-}3$ | 6.94 |
| Discrete | 1.0 | 2.0 | Yes | $5e{-}3$ | 6.08 |
| SinCos | 1.0 | 5.0 | Yes | $5e{-}3$ | 6.04 |
| SinCos | 0.5 | 5.0 | Yes | $5e{-}3$ | 6.04 |
| Discrete | 0.5 | 5.0 | No | $5e{-}3$ | 6.04 |
| SinCos | 1.0 | 5.0 | No | $5e{-}3$ | 6.04 |
| Discrete | 1.0 | 5.0 | No | $5e{-}3$ | 6.04 |
| SinCos | 0.5 | 5.0 | No | $5e{-}3$ | 6.04 |
| Discrete | 1.0 | 5.0 | Yes | $5e{-}3$ | 6.04 |
| Discrete | 0.5 | 5.0 | Yes | $5e{-}3$ | 6.04 |

Table 8: WIRE grid search combinations.

| Embedding | $w_0^{\text{initial}}$ | $w_0^{\text{hidden}}$ | Learning rate | $f$ PSNR |
|---|---|---|---|---|
| Discrete | 0.5 | 2.0 | $1e{-}2$ | 29.33 |
| Discrete | 0.1 | 2.0 | $1e{-}2$ | 27.96 |
| Discrete | 0.5 | 0.5 | $1e{-}2$ | 27.9 |
| Discrete | 0.1 | 0.5 | $1e{-}2$ | 27.83 |
| Linear | 0.1 | 2.0 | $1e{-}2$ | 24.16 |
| Linear | 0.1 | 5.0 | $1e{-}2$ | 24.16 |
| Linear | 0.1 | 0.5 | $1e{-}2$ | 24.16 |
| Linear | 0.5 | 0.5 | $1e{-}2$ | 24.16 |
| Linear | 0.5 | 2.0 | $1e{-}2$ | 24.16 |
| Linear | 0.5 | 5.0 | $1e{-}2$ | 24.16 |
| Discrete | 1.0 | 0.5 | $1e{-}2$ | 7.65 |
| Discrete | 1.0 | 2.0 | $1e{-}2$ | 7.34 |
| Linear | 1.0 | 0.5 | $1e{-}2$ | 6.04 |
| Linear | 1.0 | 2.0 | $1e{-}2$ | 6.04 |
| Linear | 1.0 | 5.0 | $1e{-}2$ | 6.04 |
| Discrete | 0.1 | 5.0 | $1e{-}2$ | nan |
| Discrete | 0.5 | 5.0 | $1e{-}2$ | nan |
| Discrete | 1.0 | 5.0 | $1e{-}2$ | nan |

## C.7 EXTRA RESULTS

Table 9: Missing metrics from Table 1. Evaluation on 60 total $f$s (10 different turbulent trajectories, six random time snapshots), sampled in the statistically steady phase. Errors in data space. Best result in bold, second best underlined.

| | Integrals $\phi$ | | Turbulence $Q^{\mathrm{spec}}, k_y^{\mathrm{spec}}$ | | |
|---|---|---|---|---|---|
| | L1($\phi$) ↓ | PC($\overline{k_y^{\mathrm{spec}}}$) ↑ | PC($\overline{Q^{\mathrm{spec}}}$) ↑ | L1($\overline{k_y^{\mathrm{spec}}}$) ↑ | L1($\overline{Q^{\mathrm{spec}}}$) ↑ |
| ZFP | 1025.50 | 0.8950 | -0.1562 | 332832.3125 | 87.3532 |
| Wavelet | 642.32 | 0.8953 | -0.9439 | 237414.7031 | 86.9227 |
| PCA | 379.48 | 0.8951 | 0.7033 | 68666.2891 | 61.5661 |
| JPEG2000 | 1627.20 | 0.8939 | -0.0161 | 801974.5000 | 86.1083 |
| NF | 79.88 | 0.9246 | **0.9727** | 2038.9197 | 45.7231 |
| PINC-NF | **18.10** | **0.9888** | 0.9660 | **56.6920** | **43.7608** |
| PINC-AE + EVA | 307.33 | 0.9520 | 0.5341 | 38401.5508 | 70.8733 |
| PINC-VQ-VAE + EVA | 39.55 | 0.9530 | 0.7334 | 251.5966 | 59.9805 |

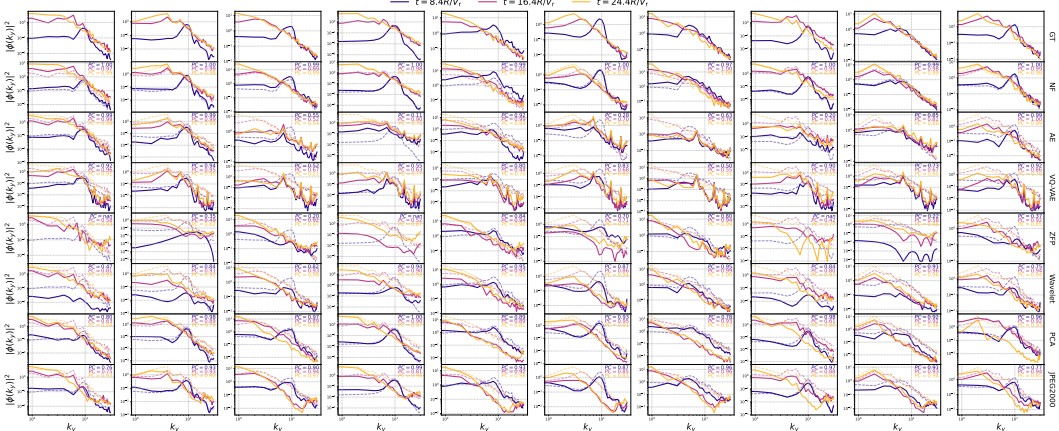

Figure 8: Extra models for the energy cascade (left Figure 5). The three time snapshots at $[8.4, 16.4, 24.4]R/V_{\mathrm{r}}$ are specifically sampled in the transitional phase where mode growth and energy cascade happens, before reaching the statistically stable phase. Visualized as the energy transfer from higher to lower modes as turbulence develops. Columns are different trajectories, rows are compression methods, lines of varied colors are the $k_y^{\mathrm{spec}}$ at specific timesteps, and transparent lines are respective ground truth.

Table 10: Timing details for neural and traditional compression, in seconds. GPU: single NVIDIA A40 (48GB), CPU: Intel Xeon Platinum 8168, 96 cores, 2.70GHz.

| Model | Offine compute | Compress [s] | Decompress [s] | Device |
|---|---|---|---|---|
| NF | - | 96.3 | 0.260 | GPU |
| AE | $\sim 4 \times 60h + 28h$ | 0.377 | 0.023 | GPU |
| VQ-VAE | $\sim 4 \times 60h + 28h$ | 0.425 | 0.027 | GPU |
| ZFP | - | 0.144 | 0.066 | CPU |
| Wavelet | - | 1.30 | 0.804 | CPU |
| PCA | - | 0.377 | 0.149 | CPU |
| JPEG2000 | - | 4.17 | 0.261 | CPU |

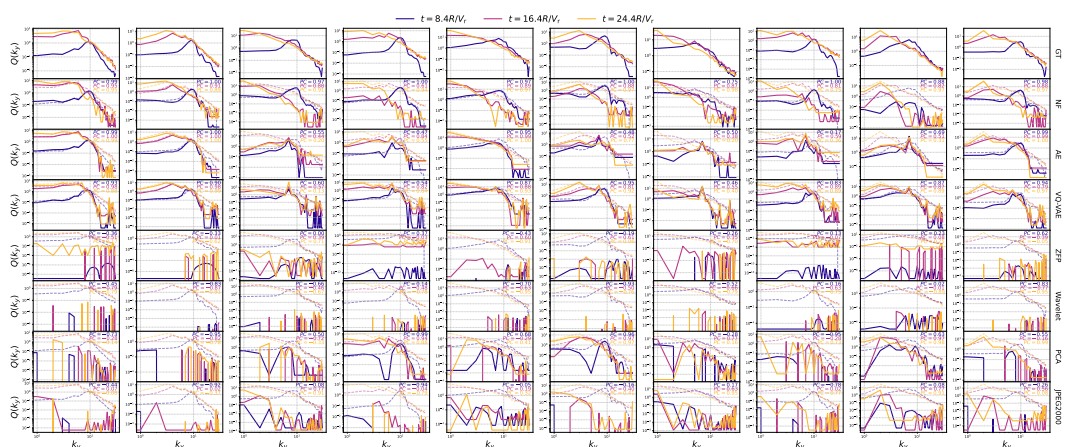

Figure 9: Extra models for the $Q$ spectra (right Figure 5). The three time snapshots at $[8.4, 16.4, 24.4] R/V_\mathrm{r}$ are specifically sampled in the transitional phase where mode growth and energy cascade happens, before reaching the statistically stable phase. Visualized as the energy transfer from higher to lower modes as turbulence develops. Columns are different trajectories, rows are compression methods, lines of varied colors are the $Q^\mathrm{spec}$ at specific timesteps, and transparent lines are respective ground truth.

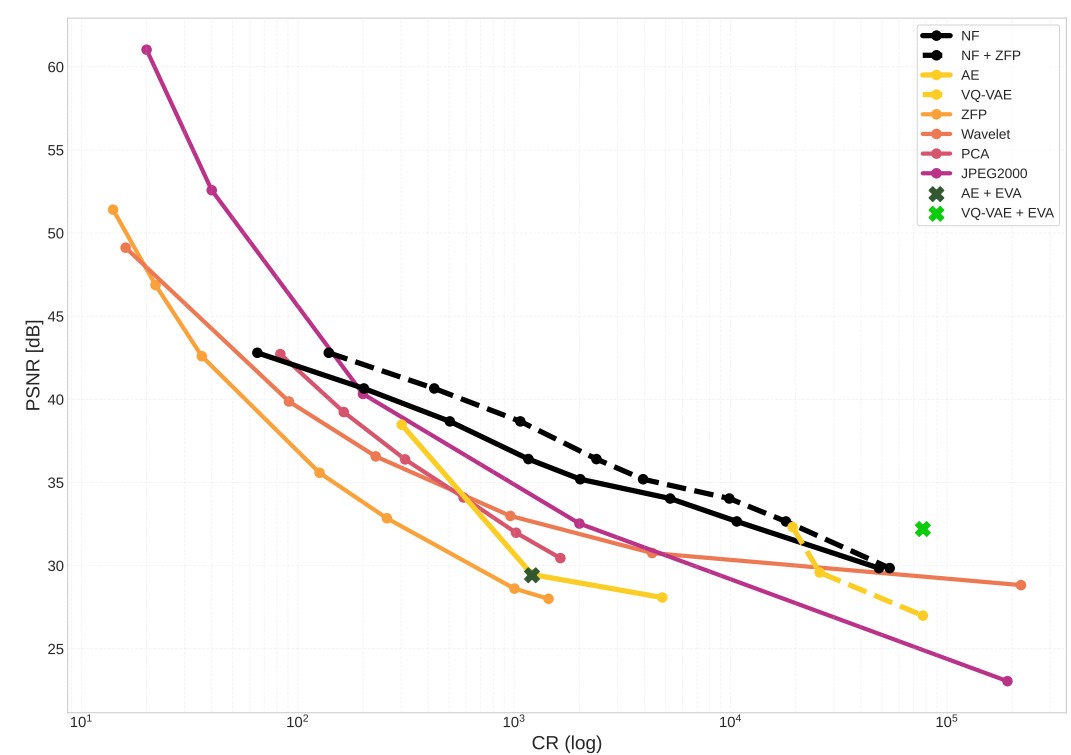

Figure 10: Full PSNR scaling plot with missing curves from Figure 2a

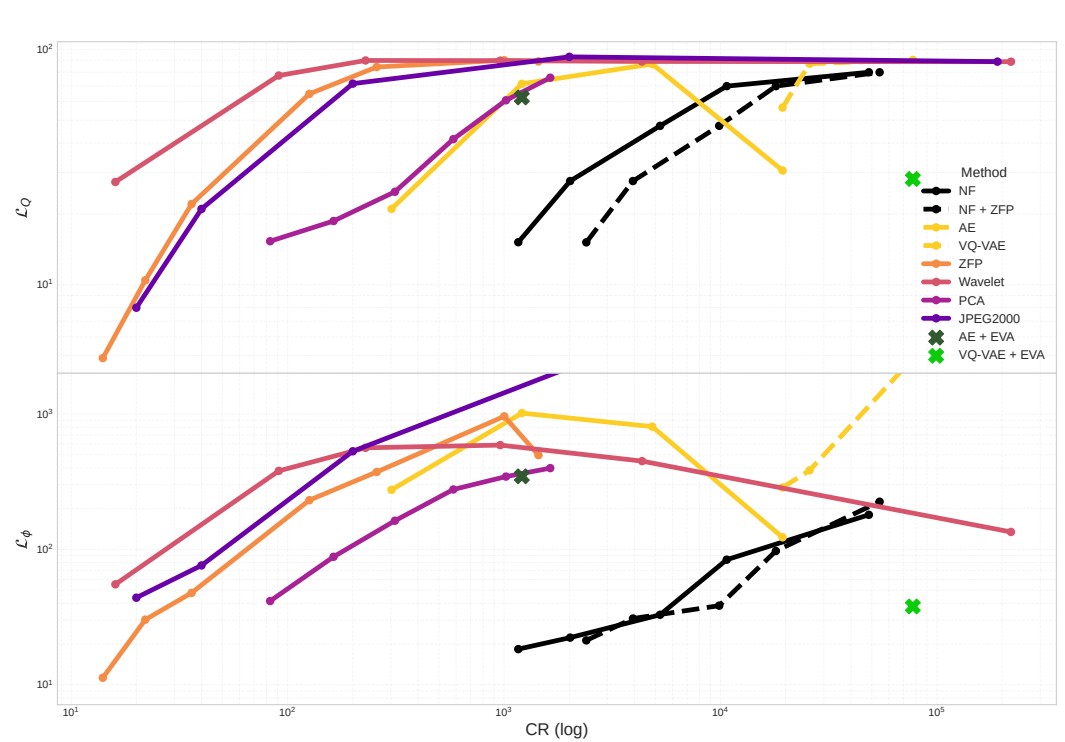

Figure 11: Full physics scaling plot with missing curves from Figure 4b

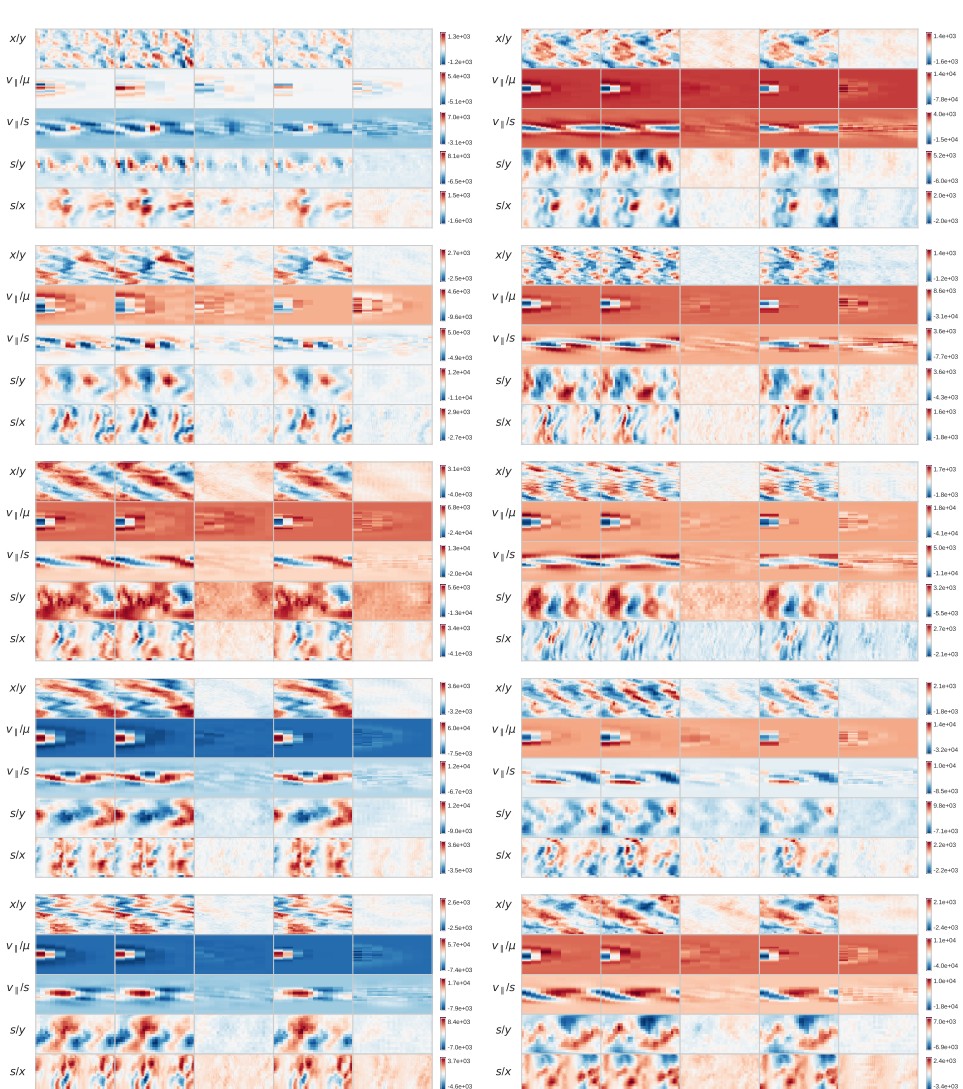

Figure 12: Extra reconstructions for the 5D density function $f$. CR $=\sim 1,000\times$. Each row is a different trajectory at timestep $176.4R/V_\mathrm{T}$. Columns match Figure 13.

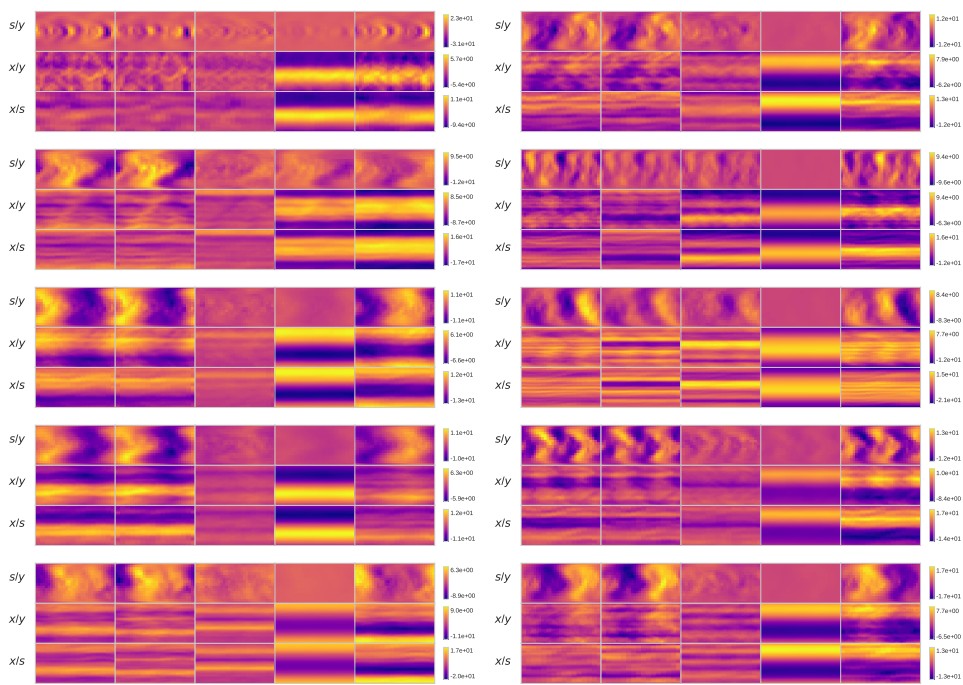

Figure 13: Extra reconstructions for the 3D electrostatic potential $\phi$. CR $=\sim 1,000\times$. Each row is a different trajectory at timestep $176.4R/V_\mathrm{r}$. Columns match Figure 12.

