# OpenReview forum: "Physics-Preserving Compression of High-Dimensional Plasma Turbulence Simulations"
_ICLR.cc/2026/Conference — Submitted to ICLR 2026_

### Official Review · Reviewer_WgAS · 2025-10-23

**Soundness:** 3
**Presentation:** 2
**Contribution:** 2
**Rating:** 4
**Confidence:** 4

**Summary:**

The paper introduces Physics-Inspired Neural Compression (PINC), a method that integrates physics-informed losses to compress massive gyrokinetic plasma simulation data by up to 70,000×. It preserves key spatial and temporal turbulence characteristics that conventional compression methods may fail to maintain.

**Strengths:**

Significance
This paper compares compression methods for high-dimensional scientific simulations, including neural field and vector quantized approaches in plasma turbulence, and presents solid experiments in a field lacking established benchmarks.

**Weaknesses:**

Originality

1. The proposed Physics-Inspired Neural Compression (PINC) lacks methodological novelty, as similar physics-informed compression frameworks have been explored in prior works such as GINN [1] and VQ-VAE for tubulence [2]. The paper mainly adopts existing  method on the plasma turbulence dataset without introducing a new architectural or algorithmic contribution.

Quality and Clarity

2. The paper does not clearly explain how the proposed physics-informed loss differs from or improves upon prior physics-guided neural compression works [1][2].

Significance

3. The experimental results (Table 1) indicate a trade-off between PSNR and L1 error when incorporating physics-informed losses, suggesting that improved physical fidelity comes at the cost of reconstruction accuracy.
4. While the evaluation pipeline is comprehensive, the method itself is incremental, and its advantages over previous approaches are not sufficiently quantified.
5. The novelty and generalizability of the proposed method are limited, which may reduce its long-term impact compared to its benchmarking contributions.

[1] Geometry-Informed Neural Networks
[2] A Physics-Informed Vector Quantized Autoencoder for Data Compression of Turbulent Flow

**Questions:**

1. Have the authors considered combining VQ-VAE with an entropy codec (e.g., arithmetic or Huffman coding) for improved hybrid compression efficiency?
2. What is the codebook utilization of the VQ-VAE component, and how does it affect compression quality and efficiency?

---

> ### Author Response · Authors · 2025-11-18
>
> We are thankful to reviewer __WgAS__ for the comments, along with interesting ideas on the VQ-VAE. In the following sections we address the raised weaknesses.
>
> ## [W1] Originality
> First, we believe there is a misunderstanding on the reviewer side: Geometry-Informed Neural Networks (Berzins et al.) are not not a "physics-informed compression framework". They are data-free generative models for topology optimization. Losses used are geometric constrains, not physics informed.
>
> Furthermore, we would like to clarify our contributions, which are two-fold:
> (1) Evaluation metrics for spatial and transient (plasma) turbulence simulations.
> (2) Tailored physics-informed losses for gyrokinetics data compression.
>
> To the best of our knowledge, neither of these have been explored in depth in prior works. Momenifar et al. investigates physics-informed losses for turbulent flow data which is not applicable to gyrokinetics: plasma turbulence exhibits unique dynamics such as a bi-directional turbulent energy cascade.
> Furthermore, some of our physics-informed losses optimize for aggregated global quantities (flux loss), which is underexplored in PINN literature and distinguishes our work from regularly employed physics losses.
>
> ---
> Berzins et al., 2025, _Geometry-Informed Neural Networks_
> Momenifar et al., 2021, _Dimension Reduced Turbulent Flow Data From Deep Vector Quantizers_
>
> ## [W2] Quality and Clarity
> We agree with the reviewer that we could make the positioning sharper, this was also raised by __Dmk1__.
> As stated in W1, our work does not improve upon the mentioned prior works, but is complementary in that we introduced evaluation metrics and training losses, particularly designed for gyrokinetics.
> We also clarified this in the Related Work section of the revision upload. These are the additions:
>
> > Contrary to the typical local, residual PINN losses, in our case they are global non-linear integrals which depend on the Fourier mode structure.
>
> > Another notable mention is Momenifar et al. (2022), which uses a physics-informed VQ-VAE to capture velocity gradients and statistical properties in isoentropic flows.
>
> ## [W3,W4,W5] Significance
> ### Ad 3: PSNR-Physics tradeoff
> Thanks for bringing this up, it is an interesting aspect of how our losses behave. We also detected this early in the project, and connected it to the specific global nature of our loss and the spectral properties of neural networks (lines 358-368).
>
> Another interpretation comes from multi-task literature. This tradeoff is documented and observed at gradient level, and called "conflict gradients" (Liu et al., 2021). The update direction that improves a physics-informed loss can be adversarial to the data loss.
>
> Conflict-free optimizers such as ConFIG (Liu et al., 2025) ensure that no loss is traded for another. We ablated it against regular Adam, however found it to weaken the physical losses at the benefit of higher PSNR (Appendix C.6, Table 4).
>
> ---
> Liu et al., 2021, _Conflict-Averse Gradient Descent for Multi-task Learning_
> Liu et al., 2025, _ConFIG: Towards Conflict-free Training of Physics Informed Neural Networks_
>
> ### Ad 4: Advantages over previous methods
> To our knowledge and expertise, prior methods are not applicable in our setting. Refer to __W1__ and __W2__ for the differences.
>
> ### Ad 5: Long-term impact
> Thanks for raising this point, which is a shared concern with __Dmk1__.
>
> We believe that our work can have long-term impact on fusion reactor design, which is currently bottlenecked by prohibitively expensive numerical simulations, one of which is gyrokinetics. Production-level gyrokinetics can take multiple days of supercompute with prohibitive storage demands, as very recently shown by Di Siena et al. for ITER gyrokinetics. Our PINC allows 1) storing those large simulations for offline analysis, and 2) training of fast, compressed machine-learned alternatives to the expensive numerics. We consider the latter direction as particularly valuable and plan it as our next step.
>
> ---
> Di Siena et al., 2025, _First global gyrokinetic profile predictions of ITER burning plasma_
>
> ## [Q1] Hybrid VQ-VAE compression
> We would like to thank the reviewer for the valuable suggestion to enhance hybrid compression. We only explored hybrid NF compression with ZipNN (lossless) and ZFP (lossy).
>
> As suggested, we include entropy coding on the VQ-VAE latents to further increase the compression ratio to 121492x (compared to the original 77656x compression ratio), extra compression of 1.56x.
>
> Method | Size (MB) | Compression Ratio
> --- | --- | ---
> Original | 723.5GB | -
> VQ-VAE (int16) | 9.32MB | 77656x
> VQ-VAE + Huffman | 5.96MB | 121492x (__+1.56x__)
>
> ## [Q2] VA-VAE codebook
> Our VQ-VAE achieves 71.4% codebook utilization (5846/8192), indicating efficient use of the discrete latent space. The sorted codebook frequencies follow Zipf's law suggesting common patterns use frequent codes, while rare events retain dedicated codes.

---

> > ### Comment · Reviewer_WgAS · 2025-11-23
> > **Response**
> >
> > 1. I do not think there exists a fundamental distinction between this work and previous works on turbulent flow compression. The rebuttal considerably overclaims the contribution and novelty of this work. Adding some physics-informed regularization is really a straightforward way to extend any data compression method.
> > 2. The marginal improvement issue and tradeoff issue for gyrokinetics have not been resolved. This is foreseeable, as it is widely known that PINNs have such issues.
> > 3. The codebook utilization is not efficient. In fact, 70% utilization is rather insufficient, which showcases that the authors do not keep up with the latest literature [1].
> >
> > Considering the above concerns, I have decided to maintain my original evaluation.
> >
> > [1] Restructuring Vector Quantization with the Rotation Trick

---

> > > ### Author Response · Authors · 2025-11-28
> > >
> > > Thank you for your engagement and the provided suggestions on related work, we address the remaining concerns below.
> > >
> > > > I do not think there exists a fundamental distinction between this work and previous works on turbulent flow compression.
> > >
> > > The fundamental difference is that we are considering a different domain of turbulence with unique characteristics (bi-directional energy cascade), not present in previously investigated flows.
> > >
> > > > The rebuttal considerably overclaims the contribution and novelty of this work.
> > >
> > > We never claimed that our contributions reach beyond the domain of gyrokinetics. As mentioned in the rebuttal, our contributions are two-fold:
> > >
> > > - Evaluation metrics for spatial and transient gyrokinetics simulations.
> > > - Tailored physics-informed losses for gyrokinetics data compression.
> > >
> > > To the best of our knowledge, none of these have been investigated in prior works.
> > >
> > > > The codebook utilization is not efficient.
> > >
> > > Thank you for pointing out the very relevant work on vector quantization. We show results for a VQ-VAE trained with the roatation trick and compared to our best VQ-VAE reported in the paper below.
> > >
> > >
> > > | Method                           | Codebook Usage        | PSNR (dB) |
> > > |----------------------------------|------------------------|-----------|
> > > | BASELINE VQ-VAE (Euclidean)      | 72.1%      | 32.32     |
> > > | VQ-VAE (Cosine) + Rotation       | 66.4%      | 31.24     |
> > > | VQ-VAE (Euclidean) + Rotation    | 59.4%      | 27.47     |
> > >
> > > In addition we experimented with other vector quantization approaches that promise improved codebook utilization [1,2]
> > >
> > > | Method                           | Codebook Usage        | PSNR (dB) |
> > > |----------------------------------|------------------------|-----------|
> > > | FSQ | 100.0% | 27.89 |
> > > | LFQ | 99.4% | 25.88 |
> > >
> > > These experiments demonstrate that (1) the rotation trick does not improve codebook utilization and (2), higher codebook utilization does not necessarily correlate with higher PSNR. We will add these new results to our manuscript.
> > >
> > > [1] Yu et al., Language Model Beats Diffusion -- Tokenizer is Key to Visual Generation, ICLR 2024
> > >
> > > [2] Mentzer et al., Finite Scalar Quantization: VQ-VAE Made Simple, ICLR 2024

---

### Official Review · Reviewer_QH9G · 2025-11-01

**Soundness:** 3
**Presentation:** 3
**Contribution:** 2
**Rating:** 4
**Confidence:** 2

**Summary:**

This paper addresses the storage and analysis bottleneck for high-fidelity 5D gyrokinetic plasma turbulence simulations, which generate terabytes of data. The authors claim that standard lossy compression methods fail to preserve essential physical characteristics, particularly transient turbulence dynamics.
The core contributions are:
1. A novel physics-informed loss function: This loss function is specifically designed for gyrokinetics and incorporates terms to preserve physical integrals heat flux, electrostatic potential, turbulence spectra, and monoticity.
2. A Proposed Evaluation Framework: The paper proposes and uses a set of metrics to evaluate both spatial/steady-state quantities (quantitatively) and transient turbulence dynamics (the latter qualitatively).
3. State-of-the-Art Compression: The PINC-VQ-VAE model achieves an extreme compression rate of 70,000x while maintaining significantly better physics fidelity than all baselines.

**Strengths:**

1. Significance: The paper tackles a real, and high-value problem in the scientific ML, where data can be incredibly high-dimensional and sparse.
2. Clever Loss Function: While the concept of physics-informed losses is well-established (e.g., PINNs), the specific formulation is a significant strength. The authors move beyond standard PDE residuals. The inclusion of losses on derived, non-local turbulence spectra—which can be difficult to compute—and the isotonic loss to enforce a physically-correct spectral shape is a non-trivial and highly effective application of this idea to the compression domain.
3. Rigorous Experiments: The evaluation is thorough and backed by an impressive 500GB dataset (although difficult to share and reproduce).

**Weaknesses:**

1. Limited Baselines: The related work section (Sec 2) mentions other relevant deep learning methods for scientific data (e.g., VAPOR, Anirudh et al., Cranganore et al.). However, the quantitative comparison in Section 4 is limited to traditional methods (ZFP, Wavelet, PCA, JPEG2000) and the authors' own non-PINC ablations. This makes it difficult to assess how PINC compares to other state-of-the-art learned compressors in this domain.
2. Overstated "Unified Evaluation Pipeline" Contribution: The paper claims to contribute a "unified evaluation pipeline". This is strong language for what is, in practice, a curated set of metrics. While valuable, it is not a new automated framework. Furthermore, the authors admit in their limitations (Line 521) that the most novel part of this "pipeline"—the evaluation of transient dynamics—remains purely "qualitative"  and is not a quantitative metric.
3. Reproducibility: In your statement, you note the dataset is too large to distribute. A far more effective solution for reproducibility would be to provide the compressed test set.

**Questions:**

1. On Baselines: Your related work review is thorough, but the experimental baselines are primarily traditional methods. Could you comment on why other learned compression methods (like VAPOR or others) were not included in the comparison?
2. On "Unified Evaluation Pipeline": In practice, you introduced a new curated set of metrics. Can this really be called a unified evaluation pipeline?
3. On Reproducibility: You state the 500GB dataset is too large to share. A much more practical solution for reproducibility would be to release the compressed test set (i.e., the latent codes), which would be negligibly small (MBs). This would allow anyone to reproduce your entire analysis pipeline (all tables and figures) without needing to re-run the GKW simulations. Would you be willing to add this to your supplementary materials?

---

> ### Author Response · Authors · 2025-11-18
>
> We wish to thank __QH9G__ for the time dedicated in writing the review and the constructive feedback. We respond to the issues and raised questions below.
>
> ## [W1/Q1] Baselines
> Thanks for raising this, we could have been clearer in the related work section.
> - ISABELA is very performant at "low" compression rates of around 7-8x, which is much lower compared to our target range where neural methods have an edge. Further, it is unclear how to increase its compression ratio and adaptation to our gyrokinetics data is cumbersome and not trivial.
> - Unfortunately, Choi et al. does not link an open-source implementation. Nevertheless, we re-implemented it and provide implementation details and results for gyrokinetics data below.
> - We already provide a comparison to the neural field techniques used in  Cranganore et al., 2025 in Appendix C.4.
>
> VAPOR uses a VQ-VAE with a sequential FNO "refiner" on the reconstruction. It is trained on $L = \alpha L_\text{recon} + \beta L_\text{VQ} + \gamma L_\text{physics}$  with $L_{\text{physics}} = L_{\text{mass}} + L_{v_{\parallel}} + L_{E_{\parallel}}$ to conserve mass, momentum and energy (parallel). As opposed to our formulation, which aims to conserve the global turbulent structure, VAPOR appears to only use a local form of the distribution function.
> VAPOR was trained on 2D (velocity space) slices instead of the global 5D view, decoupling dimensions that have strong structural dependencies. We reimplement the overall VAPOR architecture of VQ-VAE + FNO refiner, and also attempt to include the losses (details in Appendix C.4).
> Update to LLM disclosure: we used Gemini 2.5 (Pro) for VAPOR re-implementation.
>
> We train VAPOR (ours) on over 22M 2D slices (subsampled due to computational constraints), with a compression rate of 64x. Larger compression ratios are unstable for a 2D compressor (32x8 grid in our case).
>
> | Model | Dim | CR | Params  | $f$ PSNR | $Q$ L1 | φ PSNR |
> | ----- | --- | -- | ------- | -------- | ------ | ----------- |
> | VAPOR (Choi et al.)| 2D | ~16x   | 1.8M + 2.4M  |  n/a |  n/a |   n/a |
> | VAPOR (ours)       | 2D  | 64x    | 1.7M + 2.1M  | 30.5 | 65.0 | -21.7 |
> | PINC-NF (ours)     | 5D  | 1167x  | n/a          | 35.6 | 3.44 |  12.5 |
> | PINC-VQ-VAE (ours) | 5D  | 77368x | 152.2M       | 32.7 | 30.6 |  7.65 |
>
> ---
> Anirudh et al., 2022, _Review of Data-Driven Plasma Science_
> Cranganore et al., 2025, _Einstein Fields: A Neural Perspective To Computational General Relativity_
> Choi et al., 2021, _Neural data compression for physics plasma simulation_
>
>
> ## [W2/Q2] Unified evaluation pipeline
> We thank the reviewer, and agree that our phrasing has been unfortunate and potentially misleading in its current form, our evaluation is mainly designed for gyrokinetics data. Therefore we removed the term _"unified"_ from the revised version.
>
> Additionally, we agree that our current qualitative evaluating of time dynamics consistency is limited. Therefore, we include two systematic measures. As suggested by reviewer __kwUE__, we included error measures for aggregated transient spectra $k_y^{\text{spec}}$ and $Q^{\text{spec}}$.
>
> Secondly, we also leverage the _EndPoint Error_ (EPE) of the optical flow field (Baker et al., 2011), commonly used in video modeling. Given two sequences of $x_1$ and $x_2$ of $N$ frames and their $i$-th flow vectors $\mathbf{F}_1^{(i)}$ and $\mathbf{F}_2^{(i)}$, EPE is the average flow gradient field error over the frames
>
> $$
> \text{EPE}(x_1, x_2) = \frac{1}{N} \sum^N_{i=1} \|\mathbf{F}_1^{(i)} - \mathbf{F}^{(i)}_2 \|^2_2.
> $$
>
> We show results for traditional compression methods as well as for our methods in the table below. The table includes "energy cascade" (EC) errors computed as accumulated Wasserstein distance on $k_y^{\text{spec}}$ and $Q^{\text{spec}}$, as well as the endpoint error of the optical flows on the distribution function $f$ (also added to the revision, Table 5b).
>
> Model | EPE ↓ | $EC_{k_y}$ ↓ | $EC_Q$ ↓
> --- | --- | --- | ---
> ZFP | 0.058 | 0.031 | 0.715
> Wavelet | 0.033 | 0.031 | 0.061
> PCA | 0.032 | 0.032 | 0.065
> JPEG2000 | 0.027 | 0.032 | 0.176
> NF | **0.017** | 0.030 | 0.029
> PINC-NF | 0.030 | **0.011** | 0.015
> PINC-AE | 0.030 | 0.028 | **0.005**
> PINC-VQ-VAE | 0.036 | _0.018_ | _0.008_
>
> ---
> Baker et al., 2011, _A Database and Evaluation Methodology for Optical Flow_
>
> ## [W3/Q3] Reproducibility
> We agree that reproducibility is an important aspect. Therefore, we release a "validation" set to obtain Table 1 results, except for the scaling laws (for which a wider range of fields/latents is needed).
> It includes full size samples, along with neural field weights and autoencoder checkpoints. This excedes the 100mb limit for the supplementary, therefore we release it on huggingface with an anonymized user at this link: https://huggingface.co/datasets/oidaman/pinc_reproducibility

---

> > ### Comment · Reviewer_QH9G · 2025-11-20
> >
> > All comments addressed. Raising score from 4 to 8.

---

### Official Review · Reviewer_Dmk1 · 2025-11-01

**Soundness:** 4
**Presentation:** 4
**Contribution:** 3
**Rating:** 8
**Confidence:** 4

**Summary:**

This paper introduces Physics-Inspired Neural Compression (PINC), a framework for compressing high-dimensional plasma turbulence simulations while preserving essential physical properties. The authors address the challenge of storing large gyrokinetic simulation data by proposing a single evaluation pipeline that measures both spatial and temporal turbulence characteristics. They investigate two neural compression paradigms (autoencoders and neural implicit field) and augment them with physics-informed loss terms derived from gyrokinetic integrals and turbulence spectra. The resulting PINC models achieve great compression rates while maintaining key physical quantities. The paper provides detailed quantitative and qualitative analyses, demonstrating that PINC significantly improves physics preservation compared to traditional compression methods.

**Strengths:**

- The paper addresses an underexplored yet highly relevant problem: data compression for large-scale physics simulations, rather than accelerating the simulations themselves. This is a practical and impactful direction, as it targets a major bottleneck in scientific computing: data storage and accessibility.

- The proposed Physics-Inspired Neural Compression (PINC) is conceptually well-motivated, bridging neural compression and physics-informed learning in a principled way. The inclusion of physically meaningful loss terms for gyrokinetics demonstrates a strong understanding of the domain.

- The experimental evaluation is extensive and well-structured, including comparisons with traditional compression methods, ablations of individual loss components, and scaling analyses across compression ratios.

- The results are compelling, showing that the proposed approach achieves extreme compression (up to 70,000x) while maintaining physically relevant quantities, a feat that existing methods fail to achieve.

- The paper is clearly written and well organized, with solid theoretical grounding and reproducibility details (including code, configurations, and dataset description).

- The authors provide a balanced discussion of limitations and outline meaningful future directions, such as incorporating temporal consistency and extending the approach to other domains.

Overall, the work sets a new benchmark for physics-preserving compression and highlights the potential of neural networks as a viable alternative to traditional methods in high-dimensional scientific data management.

**Weaknesses:**

Paper is really strong, minor potential issues:

- While the proposed PINC framework is convincing, it remains domain-specific, with the physics-informed losses tailored to gyrokinetic equations. It is unclear how easily the method generalizes to other scientific domains (e.g., fluid dynamics, astrophysics).

- Are there insights from this study that could inform the design of neural operators or surrogate models, possibly using compressed representations as priors or initialization?

**Questions:**

- How sensitive is the proposed physics-inspired loss formulation to the specific weighting or scaling of the individual components?

---

> ### Author Response · Authors · 2025-11-18
>
> We are grateful to for the positive feedback and the recognized relevance of the problem we aim to address. We address the concerns and questions raised as follows.
>
> ## [W1] Generalizability
> We appreciate the acknowledgement that PINC is convincing and agree that it is indeed tailored to gyrokinetics data.
> In our view, this is unavoidable as for each domain there are different quantities of interest, for example our heat flux, which is a globally integrated quantitiy.
> For other domains, such as CFD, an analogous concept would be drag/lift coefficients, which require a different loss formulation.
> Other prime examples for domain-specific losses for turbulent flows and general relativity are Momenifar et al. and Cranganore et al., respectively.
> We believe that especially in the domain of AI for science, it is not straightforward to define a generalizable compression framework across domains.
> There are, however, aspects of our loss terms that could potentially transfer to other domains, for example the monotonicity loss of the turbulence spetrum.
> Furthermore, some of the newly added temporal evaluation metrics may be usefule for other domains as well.
>
> ---
> Momenifar et al., 2021, _Dimension Reduced Turbulent Flow Data From Deep Vector Quantizers_
> Cranganore et al., 2025, _Einstein Fields: A Neural Perspective To Computational General Relativity_
>
> ## [W2] Informed design of surrogate models
> This is a very intriguing line of thought and we are indeed currently exploring this direction. In our view, machine learning in such data intensive regimes could be reliably applied in two ways:
> - __Physics-informed surrogates__. First we believe our loss functions could be used as additional loss terms for existing works training physics-informed surrogates for gyrokinetics (Paischer et al.)
> - __Learning in compressed space__. To accelerate training and allow for a more diverse dataset, data can be compressed in-situ and a surrogate can be trained offline on the latent dataset. An orthogonal direction is __in-transit__ learning (Kelling et al.), where a model is trained on streaming data, as it is generated. Both approaches have drawbacks and benefits, e.g. "offline" compressed learning may exhibit error accumulation (compression error + prediction error), while in-transit learning can be inefficient since data is not explicitly stored and samples are see once. Moreover, these two approaches can be also combined, so that data gets compressed on the fly, allowing for a small "replay buffer" of relevant (latent) samples, which the surrogate model can "exploit".
>
> As for the _initialization/prior_ point, a latent model trained on compressed snapshots could be used as starting point for a second, "full" model finetuned on full-resolution samples. This also goes in the practically relevant direction of transfer learning from relatively cheap, low-fidelity gyrokinetics to extremely expensive (and impossible to store) production-level simulations.
>
> ---
> J Kelling et al. 2025, _The Artificial Scientist: in-transit Machine Learning of Plasma Simulations_
> F. Paischer et al. 2025, _GyroSwin: 5D Surrogates for Gyrokinetic Plasma Turbulence Simulations_
>
> ## [Q1] How sensitive is the PINC loss to the scales of different components?
> Loss sensitivity is a legitimate concern, and was also raised as a question by reviwer __kwUE__.
> Magnitudes vary between each loss component: for instance, the heat flux $Q$ is generally a large scalar, as it results from the summation over a 5D field. This makes the flux loss one order of magnitude larger than the others (for reference, Q: μ=105.7, σ=42.6$, φ : μ=-2.7, σ=56.7).
>
> Surprisingly, NF optimization is rather stable, even without special multi-objective optimizers or tweaking of loss weights. We also experimented with ConFIG (Liu et al., 2024) to mitigate conflicting gradients specifically designed for PINNs (Appendix C.6, Table 4), with no observed improvements. Therefore, we set all loss-scaling weights to 1 for NF training.
> On the other hand, minimizing $\mathcal{L}_{\text{PINC}}$ for autoencoders proved to be trickier, which resulted in our two-stage training, where PINC losses are trained via LoRA adapters after pretraining on f (_PINC-autoencoders_ paragraph, lines 224-227 and Appendix C.3).

---

> > ### Comment · Reviewer_Dmk1 · 2025-11-24
> >
> > Thank you for the reply! I'm satisfied with the authors rebuttal and I've decided to keep my score.

---

### Official Review · Reviewer_kwUE · 2025-11-02

**Soundness:** 3
**Presentation:** 3
**Contribution:** 3
**Rating:** 6
**Confidence:** 3

**Summary:**

This paper addresses the critical data storage and analysis bottleneck in high-fidelity scientific computing, specifically focusing on 5D gyrokinetic plasma turbulence simulations. A framework Physics-Inspired Neural Compression (PINC) is proposed to integrate physics-informed losses into neural fields and vector-quantized autoencoders. The core of PINC is a composite loss function, which penalizes deviations in key physical integrals and turbulence diagnostics. The authors also introduce a unified evaluation pipeline to assess both spatial and transient dynamics. Results demonstrate that PINC models outperform traditional baselines and standard neural models in preserving these physical metrics, achieving extreme compression ratios of up to 70,000x.

**Strengths:**

1.This paper seems to be novel and valuable. The paper tackles a highly relevant problem for the HPC and computational science communities. The core idea of applying physics-informed losses not just for solving PDEs, but for data compression.
2.The authors provide a strong empirical evaluation. The analysis in Section 4.2 and Figure 5 moves beyond static, time-averaged metrics.
3.The ablation study is conducted to validate the contribution of each module in the proposed framework effectively and the visualization shows the comparison of compression effects.

**Weaknesses:**

1.In Section 3.2, the paper explores two distinct neural compression paradigms: Neural Fields and Autoencoders. What are the respective advantages and disadvantages (e.g., encoding/decoding speed, precise storage costs, generalization) of these methods in a real-world application?
2.In Section 4.2, the analysis of the energy cascade in Figure 5 is qualitative. Why was a quantitative transient metric not used in the evaluation? For example, computing the spectral error (such as L1 or Wasserstein Distance) at each individual timestep during the transient phase and reporting its mean would seem to more directly and rigorously validate PINC's advantage in preserving transient dynamics.
3.The composite loss in Equation 6 contains six distinct terms, which suggests a heavy optimization burden. The stability of this multi-term optimization and its sensitivity to the relative weighting of these components are not sufficiently addressed.
4.The chosen baselines, while standard, are somewhat outdated. The paper would be strengthened by comparing against more recent, state-of-the-art learned compression methods.
5.In Equation 3, the paper lacks an explanation for the meanings of the symbols $\mathfrak{R}$ and $\mathfrak{S}$.

**Questions:**

See Weaknesses. The authors are strongly recommended to clarify the effectiveness of different tasks and applications of the proposed method.

---

> ### Author Response · Authors · 2025-11-18
>
> We thank reviwer __kwUE__ for the positive and detailed comments. We elaborate on the main weaknesses and questions as follows.
>
> ## [W1] Autoencoders vs neural fields
> - For autoencoders (AE), a single model is trained to generalize across simulations. AEs offer cheap compression and decompression, but require expensive “offline” training and a diverse enough training set to enable out-of-distribution generalization. Compression is explicit in latent space, and they are not discretization invariant by default.
> - Neural fields (NF) represent datapoints implicitly within network weights. Contrary to AEs, a small network is fit on each sample, meaning that there is no notion of OOD. Compression is expensive as it occurs during training. NFs are resolution invariant by design (up to a limit).
>
> The "winner" in practice depends on data generation. In our case, AEs are limited due to the dataset requirement, while compression cost of NFs is negligible compared to the gyrokinetics code, so they are a sensible choice. However, in other domains with faster numerics NFs may not be ideal.
>
> ## [W2] Quantitative temporal consistency
> We thank the reviewer for the suggestion, and notice this point was also raised by __QH9G__.
> Evaluating time consistency is an important part of our paper. We agree that aggregating transient spectra errors would be a good proxy for turbulence preservation. On top of this, we also propose to use _EndPoint Error_ (EPE) of the optical flow field (Baker et al., 2011), commonly used for evaluation of video modeling. Let $x_1$ and $x_2$ be sequences of $N$ frames and their $i$-th _flow vectors_ $\mathbf{F}_1^{(i)}$ and $\mathbf{F}_2^{(i)}$, EPE is the flow gradient field error, averaged over time
>
> $$
> \text{EPE}(x_1, x_2) = \frac{1}{N} \sum^N_{i=1} \|\mathbf{F}_1^{(i)} - \mathbf{F}^{(i)}_2 \|^2_2.
> $$
>
> We show the suggested metric for the temporal evolution of the "energy cascade" (EC) using Wasserstein distances, as well as the distribution function EPE in the table below. Results to the revised version in Table 5b.
>
> Model | EPE ↓ | $EC_{k_y}$ ↓ | $EC_Q$ ↓
> --- | --- | --- | ---
> ZFP | 0.058 | 0.031 | 0.715
> Wavelet | 0.033 | 0.031 | 0.061
> PCA | 0.032 | 0.032 | 0.065
> JPEG2000 | 0.027 | 0.032 | 0.176
> NF | **0.017** | 0.030 | 0.029
> PINC-NF | 0.030 | **0.011** | 0.015
> PINC-AE | 0.030 | 0.028 | **0.005**
> PINC-VQ-VAE | 0.036 | 0.018 | 0.008
>
> The samples used come from the transition phase at the onset of turbulence (finer version of Figure 5a). Traditional compression is comparable on EPE, our method shines in capturing turbulence and associated heat transport.
>
> ---
> Baker et al., 2011, _A Database and Evaluation Methodology for Optical Flow_
>
> ## [W3] Training stability
> Training stability is a legitimate concern, and was also raised by __Dmk1__. We observed that loss magnitudes can vary between each loss component: for instance, Q is a large scalar (for reference, Q: μ=105.7, σ=42.6$, φ: μ=-2.7, σ=56.7).
>
> NF optimization is rather stable without multi-objective optimizers or tweaking of loss weights. We experimented with ConFIG (Liu et al., 2024) in Appendix C.6 Table 4, with no observed improvements. Therefore, we set all loss-scaling weights to 1 for NFs.
>
> Optimization for AEs proved to be trickier, which led to our two-stage training via LoRA adapters. To counteract the differences in magnitude, especially in early fine-tuning, we set the $f$ loss-weight to 10.
>
> ## [W4] Baselines
> We believe that it is important to evaluate enstablished algorithms backed by different fields of information theory. For example ZFP is an advanced block quantization method, JPEG and WFT are frequency based.
>
> We agree that MLPs and autoencoders are widespread, which underpins their relevance. We chose them as baselines to showcase the impact of PINC losses. Our findings show that simple methods, like MLP-based NFs, can outperform enstablished traditional techniques if trained correctly. On this note, Appendix C.4 is an architecture search among popular NF techniques (SIREN, WIRE, etc), and we select the best performer as our base NF.
>
> In addition we added VAPOR (Choi et al.) to our revision, as it is designed for plasma physics data. We compare it to our PINC-VQ-VAE here.
>
> Model | CR | f PSNR | Q L1 | φ PSNR
> --- | -- | --- | --- | ---
> VAPOR | 64x | 30.5 | 65.0 | -21.7
> PINC-VQ-VAE | 77368x | 32.7 | 30.6 | 7.65
>
> Recent neural compression methods, such as COIN++ (Dupont et al.), or neural video compression (Zhang et al.) are not out-of-the-box applicable or scalable to high-dimensional scientific data. Extending these approaches to gyrokinetics is a fruitful avenue for future research.
>
> ---
> Dupont et al., 2022, _COIN++: Neural Compression Across Modalities_
> Zhang et al., 2025, _Learned Rate Control for Frame-Level Adaptive Neural Video Compression via Dynamic Neural Network_
>
> ## [W5] Eq. 3
> Given a complex number $x$, $\Re(x)$ and $\Im(x)$ are real and imaginary parts of $x$.

---

### Author Response · Authors · 2025-11-18

We wish to thank all reviewers for their efforts in providing constructive feedback which enabled us to significantly improve the quality of our manuscript. __Additions to the revised version are marked in green, with the respective reviewer tags in square brackets.__

We summarize the common strengths and concerns raised by reviewers, and elaborate on how we addressed the concerns.

## Common strengths
All reviewers mentioned how PINC addresses a pressing and unaddressed storage bottleneck in real world scientific computing, following a principled approach, bridging neural compression and physics-informed learning (__kwUE__, __Dmk1__, __QH9G__). Additionally, performance was also listed as a strength (__kwUE__, __Dmk1__, __QH9G__), with extreme compression ratios (up to 70,000x) while preserving physics. The strong and comprehensive empirical evaluation, with extensive ablations, scaling and a large-scale dataset was also praised (__kwUE__, __Dmk1__).

## Common concerns
__Temporal Consistency Evaluation__. Quantitative metrics assessing transient dynamics are missing (__kwUE__, __QH9G__). To address this, we add the metric suggested by __kwUE__, as well as the endpoint error of optical flow, commonly used for temporal evaluation in video processing and generation.

__Generalizability__. PINC is _gyrokinetics-specific_, and extension to other scientific fields is unclear (__Dmk1__, __WgAS__). We clarified our contributions, some of which may be transferrable to other domains, such as specific loss terms of evaluation metrics.

__Stability__. Concerns regarding the sensitivity of the various loss terms and the difficulty of the optimization (__kwUE__, __Dmk1__). We did not experience severe instabilities during optimization, especially for neural fields where equal weighting across loss terms performs well. In addition we experimented with established methods tackling conflicting gradients in Appendix C.5 and C.6.

__Related works and positioning__. Differences to prior work are unclear (__Dmk1__, __kwUE__, __WgAS__). We clarify our positioning in the respective responses and in the revised manuscript.

__Baselines__. Limited to traditional methods (e.g., ZFP, JPEG2000), no comparisons to learned compression methods (__kwUE__, __QH9G__). We reimplemented  VAPOR (Choi et al.) and added it to our manuscript. Our neural compression techniques trained with PINC significantly outperform it w.r.t. both compression rate and physics preservation.

---
Choi et al., 2021, _Neural data compression for physics plasma simulation_

---

### Comment · Area_Chair_jWD6 · 2025-11-26
**Please Review Author Response**

Dear Reviewers,

The authors have now responded to your comments. Could you review their response as soon as possible? If you have any further questions or concerns, please raise them as well.

Best,

Your AC

---

### Author Response · Authors · 2025-12-02
**On the OpenReview situation**

Dear AC, SAC, PC.
We are very sorry about the unfortunate OpenReview situation, and we fully align with the handling of the incident. We understand that this places additional responsibilities on the newly appointed Area Chairs, and we sincerely thank you for your time and effort.

We want to summarize our (biased) view of the review process. Our paper got reviews 8644 (4324). Before the discussion period was frozen, we provided additional experiments and answered all questions. In particular, we add VAPOR as a learned baseline, include two temporal metrics to verify compression consistency, and ablate different types of vector quantizations / hybrid compression.

Reviewer __QH9G__ found the additions so important that they were willing to raise their score from 4 to 8 ("All comments addressed. Raising score from 4 to 8.", see discussion). Unfortunately, we didn't have the chance to engage in further discussions with reviewer __kwUE__. We believe the process significantly strengthened our submission nevertheless. Thank you again for taking action under these unfortunate circumstances.

Best,

The Authors

---

### Meta-Review · Area_Chair_C6eA · 2025-12-20

**Summary:**

This paper introduces physics-informed neural compression to address storage bottlenecks in gyrokinetic plasma simulations. By augmenting neural fields and VQ-VAEs with domain-specific losses, the method achieves high compression rates better than traditional and learned baselines in preserving physical quantities.

The AC acknowledges the high praise from reviewer Dmk1, who notes the paper's impressive empirical performance. However, the AC remains aligned with the concerns regarding methodological novelty. As reviewer WgAS pointed out, the core technical contribution of integrating domain-specific physics losses into VQ-VAEs and neural fields is an established practice in scientific machine learning.

Consequently, the AC recommendation is rejection.
While the paper represents a high-quality application of ML to plasma physics, the AC believes that the algorithmic innovation is primarily an incremental extension of existing SciML techniques. The strengths identified by the positive reviewers reflect a successful engineering effort in a specialized domain, rather than a fundamental advancement offering broad ML insights.

**Reviewer Concerns:**

The authors addressed several concerns including comparison against learned methods (reviewer kwUE and QH9G), and lack of quantitative metrics for transient dynamics (kwUE). However, the concern regarding methodological novelty raised by reviewer WgAS remains outstanding.

**Reviewer Scores:**

Reviewers QH9G and kwUE would likely raise their scores given their primary requests for baselines and metrics were directly addressed. Reviewer WgAS explicitly maintained their score of 4 due to insufficient novelty for ICLR consideration. The AC aligns with this perspective.

---

### Decision · Program_Chairs · 2026-01-26

Reject